# Tuneable ion selectivity in vermiculite membranes intercalated with unexchangeable ions

Zhuang Liu [1,2,9] ✉, Yumei Tan[1,9], Jianhao Qian [3,4,9], Min Cao[1], Eli Hoenig[5], Guowei Yang[6], Fengchao Wang [3] ✉, Francois M. Peeters [7,8], Yi-Chao Zou [6] ✉, Liang-Yin Chu [1,2] ✉ & Marcelo Lozada-Hidalgo [5] ✉

Membranes selective to ions of the same charge are increasingly sought for water waste processing and valuable element recovery. However, while narrow channels are known to be essential, other membrane parameters remain difficult to identify and control. Here we show that $Zr^{4+}$, $Sn^{4+}$, $Ir^{4+}$, and $La^{3+}$ ions intercalated into vermiculite laminate membranes become effectively unexchangeable, creating stable channels, one to two water layers wide, that exhibit robust and tuneable ion selectivity. Ion permeability in these membranes spans five orders of magnitude, following a trend dictated by the ions' Gibbs free energy of hydration. Unexpectedly, different intercalated ions lead to two distinct monovalent ion selectivity sequences, despite producing channels of identical width. The selectivity instead correlates with the membranes' stiffness and the entropy of hydration of the intercalated ions. These results introduce an ion selectivity mechanism driven by entropic and mechanical effects, beyond classical size and charge exclusion.

Evidence from biological channels[1,2] and artificial systems[1,3,4], such as nanotubes[5–7], nanochannels[8,9], and two-dimensional (2D) nanopores[10–12] has shown that achieving selectivity between ions of the same charge requires membranes with precisely controlled pore size, typically in the subnanometre range[1]. This has attracted attention to laminate membranes made with 2D materials like graphene oxide (GO)[13–15], $MoS_2$[16] or MXenes[17–20] as systems that could enable such selectivity with scalable technologies. In these membranes, the stacked 2D crystals form narrow channels that can in principle provide such selectivity. However, these membranes swell in water, which increases the interlayer space beyond the hydrated diameters of ions in common salts, making them non-selective to these ions[13,15]. This led to the design of mechanically confined GO membranes[15], and more recently, to cation control of the interlayer spacing in GO[14] and MXenes[19,20]. In this

latter strategy, ions such as $Al^{3+}$ or $K^+$ are pre-intercalated into the membranes, and because of their strong binding to the 2D material, prevent the membrane from swelling.

In this context, clays merit attention. This abundant material can be exfoliated into 2D layers, which enables fabricating laminate membranes[21–24] similar to GO or MXene. The material consists of negatively charged aluminosilicate layers with native cations such as $Mg^{2+}$ adsorbed in the interlayer spaces that hold the layers together[21,25,26]. Crucially, these native ions can be easily substituted for others via ion exchange[27–30], which enables tuning the interlayer separation[25,26,31]. This versatility, however, is also the biggest hinderance to using vermiculite membranes in ion sieving applications. Exposing the membranes to electrolyte solutions exchanges the interlayer ions for those in the solution within hours[28] making their

[1]School of Chemical Engineering, Sichuan University, Chengdu, Sichuan, PR China. [2]National Key Laboratory of Advanced Polymer Materials, Sichuan University, Chengdu, Sichuan, PR China. [3]Department of Modern Mechanics, University of Science and Technology of China, Hefei, PR China. [4]Department of Civil and Environmental Engineering, Rice University, Houston, TX, USA. [5]Department of Physics and Astronomy, The University of Manchester, Manchester, UK. [6]School of Materials Science and Engineering, Sun Yat-sen University, Guangzhou, PR China. [7]Departamento de Fisica, Universidade Federal do Ceara, Fortaleza, Brazil. [8]Department Physics, University of Antwerp, Antwerpen, Belgium. [9]These authors contributed equally: Zhuang Liu, Yumei Tan, Jianhao Qian. ✉e-mail: liuz@scu.edu.cn; wangfc@ustc.edu.cn; zouych5@mail.sysu.edu.cn; chuly@scu.edu.cn; marcelo.lozadahidalgo@manchester.ac.uk

properties unstable and sometimes even leading to complete membrane disintegration. One strategy to tackle this problem is to chemically bond the layers with organic or oxide linkers, which preserves membrane stability and enables ion-selective membranes[32–34]. In this work, we adopt a different strategy. We show that a group of high valence ions, namely zirconium, iridium, tin and lanthanum intercalated in vermiculite clay laminates, is effectively impossible to exchange at ambient conditions yielding robust and tuneable ion selective membranes.

## Results

### Membranes with robust ion-exchange resistance

Vermiculite laminate membranes were fabricated from aqueous dispersions prepared from thermally expanded vermiculite crystals via a two-step $Na^+$ and $Li^+$ ion exchange process, as reported previously[21–23]. This yielded stable aqueous colloid solutions of high-quality vermiculite nanosheets ~1 nm thick and ~10 μm in lateral size (Fig. 1a and Supplementary Fig. 1). Membranes with a well-ordered lamellar structure, typically ≈2 μm thick (Fig. 1c), were fabricated from the dispersions via vacuum filtration (Fig. 1a, Supplementary Fig. 1). This as-fabricated vermiculite membrane, saturated with $Li^+$ ions (Li-Ver), was then immersed in different electrolyte solutions to exchange the $Li^+$ cations for those in the electrolyte. We tested various cations, such as $K^+$, $Na^+$ or $Cs^+$, all of which were successfully inserted into the membrane, yielding K-Ver, Na-Ver, etc. However, while this makes vermiculite membranes highly tuneable, it also makes them unstable as ion separation membranes. This can be clearly appreciated for $Li^+$ electrolytes, for which the problem is particularly severe. We found that the sharp XRD signal of these membranes (e.g. K-Ver) is lost within ~24 h of continuous exposure to LiCl electrolyte (Fig. 1d inset) and visual examination revealed that these membranes had disintegrated (Supplementary Fig. 2). For other electrolytes (e.g. K-Ver in NaCl), the membranes did not disintegrate in the electrolyte, but their composition changed rapidly. Inductively coupled plasma optical emission spectrometry (ICP-OES) of the membranes revealed that within 24 h of exposure to different electrolyte solutions, the mass loss of interlayer ions was >95%; that is, practically all the initial intercalated ions were exchanged (Fig. 1e inset and Supplementary Fig. 3).

We found that $Zr^{4+}$, $Ir^{4+}$, $Sn^{4+}$ and $La^{3+}$ inserted in vermiculite membranes are effectively impossible to remove by immersing the membranes in common electrolytes at ambient conditions and that this makes these membranes robust in ion transport experiments. To fabricate the membranes, the Li-Ver membranes described above were immersed in electrolyte solutions of these ions (e.g. 1.0 M $ZrCl_4$ for about 1 h) and then rinsed with deionised (DI) water. We then tested the stability of the exchanged membranes by immersing them in various aqueous electrolyte solutions, including LiCl, between 1 to 180 days. XRD characterisation showed that the interlayer separation of the membranes remained stable for as long as we decided to test them, up to 180 days (Fig. 1d and Supplementary Figs. 4, 5). Additional ICP-OES characterisation revealed only a minor ~5% loss of intercalated ions (Fig. 1e, Supplementary Fig. 5), which remained stable over the

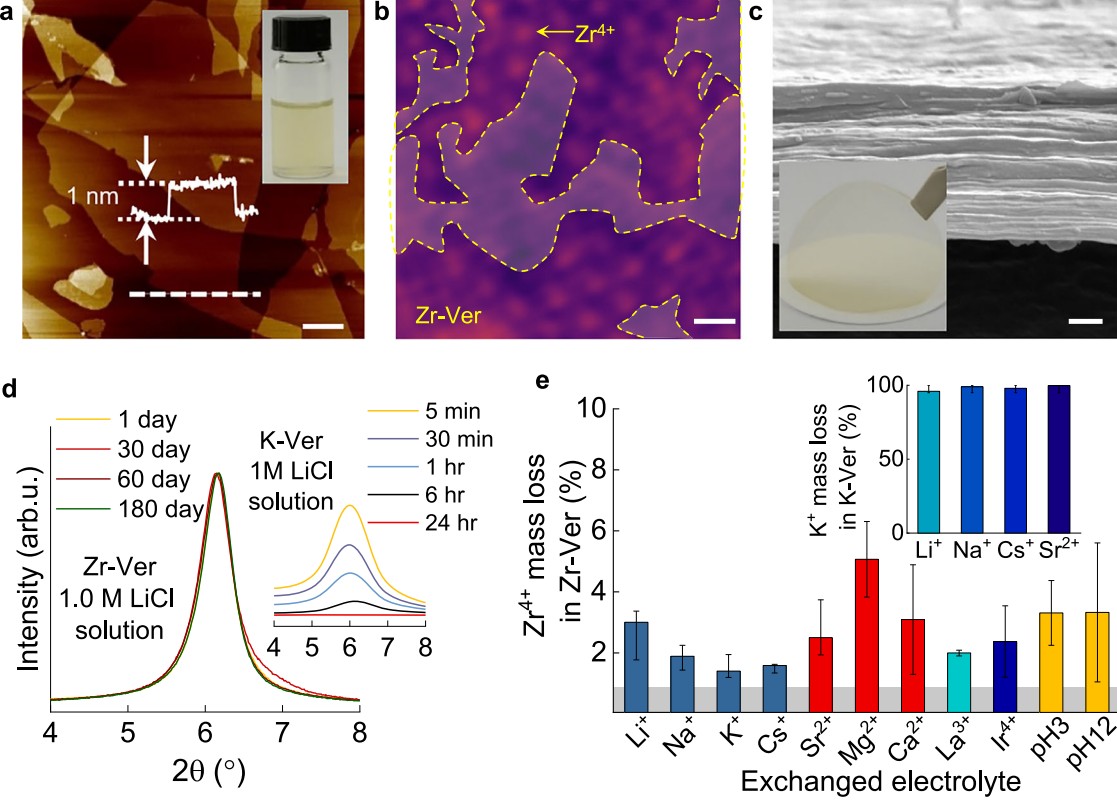

**Fig. 1 | Robust ion-exchange resistance of Zr-intercalated vermiculite membranes. a** Atomic force microscopy image of exfoliated vermiculite flakes. Bottom inset, height profile of crystal along section marked by dashed line below. Scale bar, 2 μm. Top inset, aqueous dispersion of the exfoliated crystal. **b** Annular dark field scanning transmission electron micrograph of Zr-intercalated (Zr-Ver) flake. The crystal consists of aluminosilicate layers with adsorbed $Zr^{4+}$ ions (marked with yellow arrow). $Zr^{4+}$ ions display bright contrast due to their relatively heavier atomic number. Yellowed dashed lines mark regions that are not occupied by $Zr^{4+}$. Scale bar, 5 Å. **c** Cross-sectional scanning electron microscopy image of a typical vermiculite membrane. Scale bar 500 nm. Inset, optical image of typical membrane. **d** X-ray diffraction (XRD) spectra of Zr-Ver membrane shows no changes after 180 days of immersion in LiCl. Inset shows that the XRD signal for $K^+$ intercalated vermiculite (K-Ver) immersed in LiCl is lost within 24 h. **e** Mass loss of $Zr^{4+}$ ions in Zr-Ver membranes determined with mass spectrometry after exposing the membrane to different chloride electrolytes, acid (HCl, pH = 3) and basic (NaOH, pH = 12) solution for 24 h. Grey band marks the experimental sensitivity. Inset, corresponding data for K-Ver. Error bars, standard deviation from different measurements.

whole testing period. We therefore call these strongly bonded ions 'unexchangeable ions', to differentiate them from those that can be readily exchanged (e.g. K⁺ or Ba²⁺, Supplementary Fig. 3). Microscopic insights into the distribution of the unchangeable ions were obtained with aberration-corrected scanning transmission electron microscopy (AC-STEM). This revealed that the ions tended to form domains with quasi-hexagonal symmetry separated by large $Zr^{4+}$-free regions (Fig. 1b). In fact, only a small fraction of the flake surface is occupied by $Zr^{4+}$, leaving extensive areas without coverage (Fig. 1b and Supplementary Fig. 1). This low coverage is expected, as the 4+ valence of each $Zr^{4+}$ ion balances multiple adsorption sites in vermiculite (each with −1 valence).

## Selective ion transport

These robust laminates could be used as ion sieving membranes. We measured the permeability of 14 common inorganic cations focusing on Zr-Ver membranes, which is used as a model system. In the experiments, the membrane was used as a separator between two reservoirs, one containing a solution of a common chloride salt and the other filled with deionised (DI) water (Fig. 2a inset)[20,35,36]. The concentration of ions in this latter reservoir was then measured as a function of time using ICP-OES and atomic absorption spectroscopy (AAS). Figure 2a shows that such concentration increased linearly with time and decreased linearly with membrane thickness, $l$ (Supplementary Fig. 6). This allowed extracting the ions' flux through the membrane, $J$, and the membrane permeability $P$ as: $P = J \, l \, \Delta C^{-1}$, with $\Delta C$ the difference in electrolyte concentration between feed and permeate chambers. These experiments revealed that the permeability of different monovalent cations extended over an order of magnitude, with Cs⁺ the most permeable ($P \sim 10^{-11} \, m^2 \, s^{-1}$) and Li⁺ the least ($P \sim 10^{-12} \, m^2 \, s^{-1}$); whereas multivalent cations all displayed lower permeability of $P \sim 10^{-12}$–$10^{-13} \, m^2 \, s^{-1}$. Reference experiments with nitrate salts yielded the same cation selectivity sequence, indicating that the selectivity is intrinsic to the membranes, and not dependent on the specific salt used (Supplementary Fig. 7). We also measured the permeability of the counterions in solutions with various salts, which revealed that all anions permeated through the membrane at a similar rate than their associated cation (Supplementary Fig. 8). This was expected. The unexchangeable $Zr^{4+}$ ions already maintain the membrane's electroneutrality and remain adsorbed during permeation experiments. Hence, anions must accompany cation permeation to preserve charge balance. No appreciable osmotic flow was observed during the measurements, and vapour transport measurements revealed low water flux (Supplementary Fig. 9), consistent with the narrow channels in the membranes and the small slip length of water in clays[37].

To understand the membrane's selectivity, we started by plotting the cation permeability versus their Gibbs energy of hydration[35] (Fig. 2e, blue symbols). This revealed that the cations' permeability was strongly correlated with their Gibbs energy of hydration, with lower hydration energy associated with larger $P$. On the other hand, XRD measurements of the membranes immersed in the different electrolytes (Fig. 2c, d and Supplementary Fig. 4) revealed that for most electrolytes, the interlayer spacing in Zr-Ver membrane was $d \approx 14.8 \, Å$. Given the vermiculite aluminosilicate monolayer thickness ($\approx 10 \, Å$), this $d$ accommodates $\approx 2$ water layers, revealing that the membrane's interlayer spacing is comparable to or smaller than the hydrated diameters of common cations. The only exceptions to $d \approx 14.8 \, Å$ were Cs⁺ and K⁺ electrolytes, which led to even narrower $d \approx 12.2 \, Å$, just one water layer thick (XRD measurements in Methods). This well-known phenomenon, so-called 'collapse' of the interlayer space, arises because Cs⁺ and K⁺ are easily dehydrated, and are stabilised as adsorbed inner sphere complexes on the lattice, forming polar bonds with the structural oxygen atoms in the vermiculite layers[31,38–41]. During interlayer collapse, the areas of the flake covered with $Zr^{4+}$ retain a fixed $\approx 14.8 \, Å$ interlayer distance. However, since areas without $Zr^{4+}$ dominate the surface, their collapse governs the overall XRD response. This results in spectra centred at $\approx 12.2 \, Å$, albeit with a broad

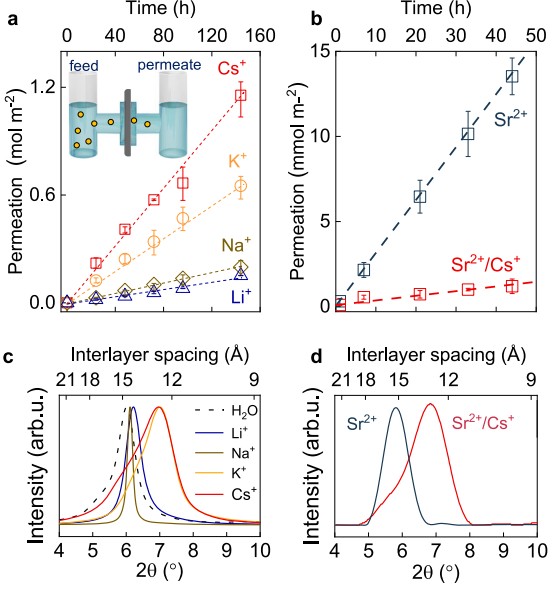

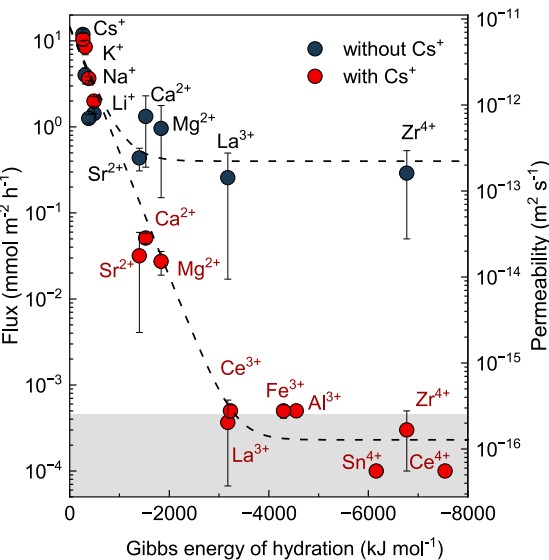

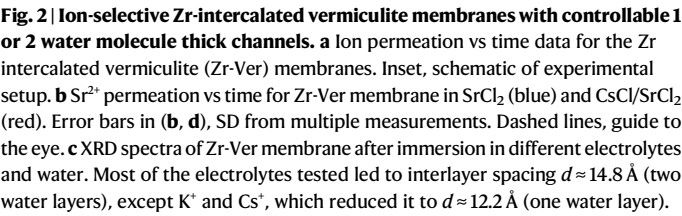

**Fig. 2 | Ion-selective Zr-intercalated vermiculite membranes with controllable 1 or 2 water molecule thick channels. a** Ion permeation vs time data for the Zr intercalated vermiculite (Zr-Ver) membranes. Inset, schematic of experimental setup. **b** Sr²⁺ permeation vs time for Zr-Ver membrane in SrCl₂ (blue) and CsCl/SrCl₂ (red). Error bars in (**b**, **d**), SD from multiple measurements. Dashed lines, guide to the eye. **c** XRD spectra of Zr-Ver membrane after immersion in different electrolytes and water. Most of the electrolytes tested led to interlayer spacing $d \approx 14.8 \, Å$ (two water layers), except K⁺ and Cs⁺, which reduced it to $d \approx 12.2 \, Å$ (one water layer). **d** X-ray diffraction (XRD) spectra of membrane after immersion in SrCl₂ (blue curve) and a 50:50 CsCl/SrCl₂ mix (red curve). Cs⁺ addition consistently reduces the interlayer spacing to $\approx 12.2 \, Å$ across all tested electrolytes. **e** Ion flux vs. Gibbs hydration energy for permeating cations (blue). Adding Cs⁺ (50:50 electrolyte-CsCl mix) leads to notably sharper ion selectivity (red). Dashed curves, guide to the eye. Right $y$-axis: ion permeability. Grey area: experimental resolution background. Error bars, standard deviation from at least ten different samples. Permeability and interlayer separation data are provided in Supplementary Table 1.

distribution due to the less uniform interlayer spacing (Fig. 2c, d and Supplementary Fig. 4).

The collapse of the interlayer space with Cs⁺ electrolyte provides a way to study ion diffusion in channels just one water molecule thick, offering further insight into the membranes' selectivity. To test this, we added 1 M of CsCl into all our 1 M chloride solutions and measured ion transport as discussed above. XRD characterisation confirmed interlayer spacing of $d \approx 12.2$ Å, corresponding to a single water layer inside the channels (Fig. 2d panel and Supplementary Fig. 4). Figure 2e shows that ion flux in these narrower channels remained correlated with their Gibbs energy of hydration, but with a much sharper dependence. Monovalent ion flux remained similar, but divalent ions now displayed $P$ about 100 times smaller than in the absence of Cs⁺ and trivalent and tetravalent ions displayed $P$ about 1000 times smaller, rendering them effectively impermeable within our measurement's sensitivity (Fig. 2b, e). As a result, ion flux now spanned five orders of magnitude. This pronounced selectivity stems from the increased energy cost associated with ion transport through a space confined to a single water layer. Taken together, the selective ion transport phenomena can be understood as arising from the difference in the ions' Gibbs energy inside and outside the channels ($\Delta G$). Weakly hydrated monovalent ions are stabilised by the channel walls and permeate easily even in the narrowest channels. This is particularly evident for Cs⁺, which is more strongly attracted to the channel walls than to water, leading to interlayer collapse and high flux. In contrast, strongly hydrated multivalent ions experience a much larger $\Delta G$, interact weakly with the walls, and display low flux.

**Tuneable monovalent ion selectivity sequences**

We then studied the transport of monovalent ions through membranes intercalated with different unexchangeable ions, namely Zr⁴⁺, La³⁺, Ir⁴⁺ and Sn⁴⁺. As shown in Fig. 3a, La-Ver and Zr-Ver membranes followed the same selectivity sequence shown above, Cs⁺ > K⁺ > Na⁺ ≳ Li⁺, whereas Ir-Ver and Sn-Ver displayed a different sequence: Li⁺ > Na⁺ ≈ Cs⁺ ≈ K⁺. As a result, $P$ for Li⁺ in Ir-Ver and Sn-Ver was about 60

times higher than in Zr-Ver and La-Ver. This behaviour remained stable for as long as we decided to test these membranes, typically >100 h (Fig. 3c, see Supplementary Fig. 10 for mixed-salt measurements). To understand this finding, and given that ion permeability depends on both the ions' equilibrium concentration and diffusion constant within the membrane[42], we measured the composition of the membranes with ICP-OES after the transport measurements (Supplementary Fig. 11). This revealed that differences in concentration of the permeating ion in the membranes explained nearly all permeability variations. This was confirmed with electrochemical impedance spectroscopy measurements (Supplementary Fig. 12), which showed that the ions' diffusion constants varied negligibly across different membranes. For example, the concentration of Li⁺ was ≈50 times higher in Sn-Ver and Ir-Ver than in Zr-Ver and La-Ver, but the diffusion constant changed by less than 20%. To understand why the concentration of the ions changed so strongly across the four different membranes, we first measured the XRD spectra of the membranes immersed in the different electrolytes. Intriguingly, this revealed that the interlayer spacing of the four membranes was the same for a given electrolyte (Fig. 3b and Supplementary Fig. 5): $d \approx 14.8$ Å, for LiCl and NaCl and $d \approx 12.2$ Å for CsCl and KCl, thus ruling out channel width differences as the cause of these different selectivity sequences.

At first glance, these results seem puzzling, since the selectivity cannot be explained by differences in channel width, nor does it correlate with the charge of the unexchangeable ions. For example, despite the different charges of Zr⁴⁺ and La³⁺, both Zr-Ver and La-Ver exhibit similar selectivity. To solve this puzzle, we draw insights from biological channels, in which ion selectivity is thought to be primarily determined by three parameters[43]: hydration free energy of the permeating ion, electrostatic interaction between ion and channel and, finally, channel elasticity. Since the first two parameters do not account for our observations, we turned our attention to channel elasticity. This parameter affects ion selectivity in biological porins because it changes the Gibbs energy of ions inside the channel due to thermal fluctuations of channel width[43]. In a first approximation, this elastic

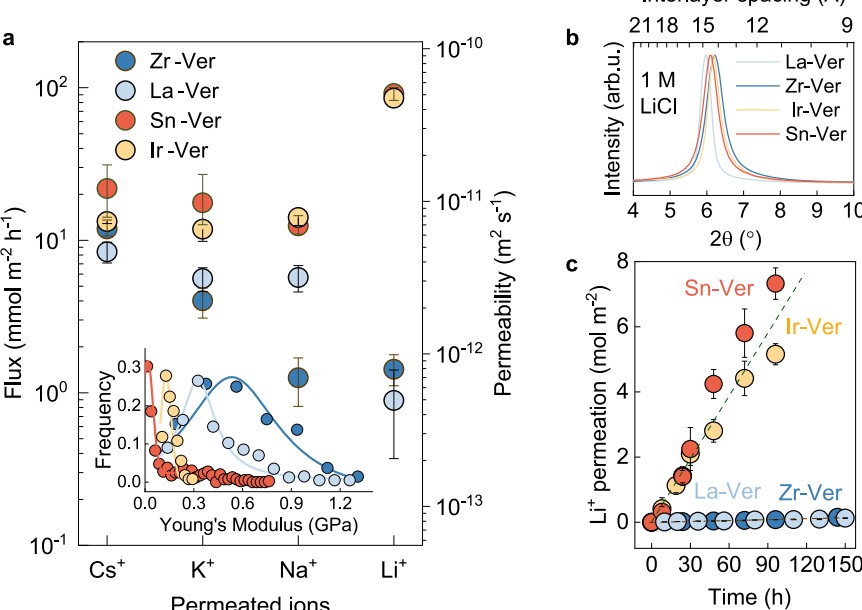

**Fig. 3 | Monovalent ion selectivity in vermiculite membranes with different intercalated ions. a** Ion permeability through vermiculite membranes intercalated with Zr⁴⁺, La³⁺, Ir⁴⁺, and Sn⁴⁺, revealing two distinct selectivity sequences. Inset, statistics of Young's Moduli of membranes obtained from atomic force microscopy measurements. **b** X-ray diffraction spectra of intercalated membranes after 24 h in

LiCl, all showing an interlayer spacing of ≈14.8 Å within experimental scatter. **c** Li⁺ permeation vs. time for membranes intercalated with different unexchangeable ions. Error bars, standard deviation from at least ten samples. Dashed lines, guide to the eye.

component is modelled as $G_e(\bar{x}) = (1/2) k(\bar{x}) [d(\bar{x}) - d_o(\bar{x})]^2$, where $k(\bar{x})$ is the elastic constant as a function of position in the channel, $\bar{x}$, and $d_o(\bar{x})$ is the equilibrium channel width[43]. Similar ionic-mechanical coupling effects have been observed in clays, with time-resolved X-ray scattering revealing that ion exchange is accompanied by dynamic fluctuations of the basal plane spacing[44]. Motivated by this, we measured the Young's moduli of the membranes, which determines the elasticity of the channels[45]. As shown in Fig. 3a (inset) and Supplementary Fig. 13, the moduli of Zr-Ver and La-Ver (0.55 GPa and 0.3 GPa, respectively) were an order of magnitude larger than for Sn-Ver (0.01 GPa) and Ir-Ver (0.03 GPa), suggesting a link between selectivity and membrane stiffness. Hence, understanding why different unexchangeable ions lead to different membrane stiffness should provide insights into the selectivity phenomena.

With this insight, we recall that water exhibits an unusually high resistance to local deformation due to its hydrogen-bond network[46,47]. This can be quantified by several thermodynamic parameters, such as cohesive energy density, isothermal compressibility, heat capacity density, and dipole orientational correlation[46]. Collectively, these quantities capture the energetic cost of disrupting its structured environment and can be used to define a quantity known as water 'stiffness', which is the work required to create a cavity for an additional water molecule[46]. Ions with highly ordered hydration shells—and thus with more negative hydration entropy, $\Delta S$—are known to lead to a more ordered water network, thus increasing water's stiffness[46,48]. Since confined water in clays is more structured than bulk[49], the influence of solvated ions on its stiffness, and thus the mechanical properties of the membrane, can be expected to be even larger. This is consistent with our observations. Of the unexchangeable ions tested, $Zr^{4+}$ ($-763\,J\,mol^{-1}\,K^{-1}$) and $La^{3+}$ ($-454\,J\,mol^{-1}\,K^{-1}$) have the most negative $\Delta S$[50] and also yield the stiffest membranes. On this basis, we propose that the unexchangeable ions modulate the stiffness of water in the membrane, thereby controlling the channels' elasticity, and thus modulating ion selectivity, similar to the role of elasticity in biological channels. This explains why $Li^+$ displays the largest variations in permeation: having the largest hydrated diameter amongst monovalent ions, it displays the largest changes in $G_e$.

## Discussion

This work investigated vermiculite laminates with immobilised high-valence ions, creating structurally stable membranes with channels either one or two water layers wide. The permeability of different cations spanned orders of magnitude and correlated with their hydration Gibbs energy. Unexpectedly, membranes with different unexchangeable ions showed different monovalent ion selectivity, which could not be explained by channel width, ion charge, or hydration Gibbs energy alone. Mechanical measurements revealed a link between membrane stiffness and the hydration entropy ($\Delta S$) of the intercalated ions, which in turn reflects the ions' impact on the structure of confined water within the membrane. This finding mirrors principles observed in biological ion channels, where entropic and mechanical effects—rather than solely size or charge exclusion – govern highly specific ion transport. Our findings establish vermiculite membranes as a versatile and robust model system to explore novel ion selectivity phenomena.

## Methods
### Materials

Vermiculite clay was sourced from Sigma Aldrich (CAS No 1318-00-9). To characterise its elemental composition, bulk crystals with grain-size of 2–3 mm were initially subjected to thermal expansion and subsequently ground into powders, dried at 80 °C for 24 h and then dissolved using dilute HF. Inductively coupled plasma atomic emission spectroscopy (ICP-AES) was used to determine the elemental composition. The detected interlayer metal cation mass fractions were 3.20%

$K^+$ ($K_2O$: 3.86%), 14.99% $Mg^{2+}$ (MgO: 24.98%), 7.77% $Al^{3+}$ ($Al_2O_3$: 14.35%), 7.77% $Fe^{3+}$ ($Fe_2O_3$: 11.08%), and 18.95% Si ($SiO_2$: 40.60%).

### Vermiculite suspension preparation

Vermiculite crystals were exfoliated in a two-step ion exchange process[21–23], as depicted in Supplementary Fig. 1. In the process, 100 mg of vermiculite granules 2–3 mm in size were immersed in 200 mL of saturated NaCl solution (36 wt%) and refluxed for 24 h. The treated vermiculite flakes were filtered and washed extensively with water and ethanol to remove residual salts. Next, 200 mL of a 2 M LiCl solution was used to disperse the sodium-exchanged vermiculite, followed by a 24-h reflux to facilitate $Li^+$ ion exchange. The resultant lithium vermiculite flakes were filtered and subjected to thorough washing with 1 L of hot water and 0.5 L of acetone. Then, the lithium-exchanged vermiculite flakes were dispersed in water and underwent three cycles of centrifugation at $1076 \times g$. This dispersion was subsequently sonicated for 20 min at 100 W to exfoliate the material into monolayer vermiculite flakes. Further centrifugation at $1076 \times g$ was employed to eliminate multilayers and bulk residues. The resulting final dispersions exhibit a pronounced Tyndall effect (Supplementary Fig. 1b), evidencing high quality dispersions.

To determine the concentration of vermiculite in the suspensions, 1 mL of the suspension was dropped onto an Anodic Aluminium Oxide (AAO) membrane with a diameter of 2.5 cm and dried. After drying, the thin film was peeled off the AAO membrane and weighed to calculate the dispersion concentration. Alternatively, the difference in mass before and after coating the AAO membrane with vermiculite suspension was also used to calibrate the dispersion concentration. Three thin film samples were typically used for concentration calibration.

We characterised the vermiculite flakes using optical microscopy, transmission electron microscopy (TEM) and atomic force microscopy (AFM). For optical microscopy observations, the vermiculite stock solution was diluted to a concentration of $10^{-4}$ mg/mL, and the diluted suspension was dropped onto a silica substrate. The sample was dried at 80 °C, followed by heating at 200 °C for 2 h. We found that most of the flakes had dimensions ranging from 10 to 20 μm (Supplementary Fig. 1c). TEM images revealed that the large vermiculite flakes exhibited no noticeable defects (Supplementary Fig. 1d).

### Membrane preparation

Vermiculite laminate membranes were prepared by vacuum filtration and ion exchange. The dispersion of exfoliated vermiculite nanosheets underwent vacuum filtration using polyether sulphone (PES) membranes (Millipore) or AAO membranes (Whatman) as filters. The PES membranes with pore size of 0.22 μm and diameter of 2.5 cm or 4.7 cm were purchased from Millipore. The AAO membranes (pore size of 0.2 μm and diameter of 2.5 cm) were purchased from Whatman. Generally, the filtration process to form the vermiculite membrane takes 1–3 days to complete. Once finished, the substrate with the vermiculite membrane is removed from the filtration apparatus and transferred to an oven for drying for 12 h. The vermiculite membranes can then be easily peeled off the substrate, resulting in intact and complete pristine vermiculite membranes.

The interlayer ions in the resulting membranes are $Li^+$ due to the exfoliation procedure. To replace them for 'unexchangeable' ions, the membranes were immersed in solutions of either 1 M of ZrCr, IrCl, SnCl or 0.1 M for IrCl. The concentration of the ion (e.g. $Zr^{4+}$) within the membranes was measured by dissolving the membranes in a 2.0 mL solution of 1 wt% dilute HF and then measuring the solution with ICP-OES. We found that the exchange process equilibrated within 1 h for 1 M solutions and over 24 h for 0.1 M solutions. Using this method, vermiculite membranes were immersed in various ion solutions to prepare the various membranes for the study (e.g. K-Ver, Sr-Ver, Ir-Ver, etc). In all cases, including $SnCl_2$, the solutions used were well-dispersed and did not display obvious precipitation. TEM

characterisation of the resulting membranes revealed that the ions, including $Sn^{4+}$, were incorporated as single atoms.

## Stability of membranes immersed in electrolyte solutions

We employed three criteria to evaluate the stability of the membranes immersed in electrolyte solutions: the membrane's visual appearance, the interlayer spacing characterised by XRD and the mass loss of the inserted ion characterised by ICP mass spectrometry.

For optical tests, $1 cm^2$ membranes were soaked in both pure water and various salt solutions, such as 1.0 M LiCl. Membranes exchanged with unexchangeable ions ($Zr^{4+}$, $Ir^{4+}$, $Sr^{4+}$ and $La^{3+}$) exhibited robust mechanical integrity. The membranes remained intact even under vigorous agitation after weeks of immersion in the electrolyte solution. In contrast, membranes intercalated with ions such as $K^+$, $Ba^{2+}$ and $Sr^{2+}$ (Supplementary Fig. 3) disintegrated after 24 h.

To test the compositional stability of the membranes, a $1 cm^2$ membrane was cut in six squares, dried at 80 °C for 24 h and weighed with an electronic balance (Mettler Toledo, XPE26; accuracy: 0.001 mg). Each membrane was then soaked in 2 mL of different electrolyte solutions. We tested 1 M LiCl, 1 M NaCl, 1 M KCl, 1 M CsCl, 1 M $SrCl_2$, 1 M $MgCl_2$, 1 M $CaCl_2$, 1 M $LaCl_3$, 0.1 M $IrCl_4$, as well as acidic (HCl, pH ≈ 3) and basic (NaOH, pH ≈ 12) solutions or pure water for 24 h. After soaking, the membranes were rinsed with deionised water, degraded with dilute HF, and analysed with ICP-OES to determine the content of the unexchangeable ion in the membrane. The membranes soaked in pure water served as a reference for total unexchangeable ion content in the original membrane. The results for ion mass loss are presented in Fig. 1e and Supplementary Fig. 5, revealing mass loss typically below 5% and always below 10%. In contrast, the mass loss in reference membranes exchanged with $K^+$, $Ba^+$ and $Sr^+$ was nearly 100% after the exposure to the electrolytes for the same period (Supplementary Fig. 3).

The interlayer spacing in the membranes in aqueous environments was characterised via X-ray diffraction (XRD) analysis (Rigaku MiniFlex 600-C with Cu Kα radiation). $1 cm^2$ membranes were soaked in 5 mL of the chloride solutions studied in this work (1 M concentration, except for $Fe^{3+}$, and $Sn^{4+}$, which were 0.1 M) for 24 h. After soaking, the membranes were thoroughly rinsed with deionised water. The wet membranes were mounted on a silicon wafer for XRD measurements, with excess water removed using filter paper. Scans were performed at 10°/minute with a step size of 0.01°. For membranes soaked in 1 M LiCl for 180 days, the most aggressive test for vermiculite membranes, no significant shift or disappearance of XRD peaks was observed. We also performed this experiment with all the electrolytes used in this study. Supplementary Figs. 4, 5 show that the XRD was stable for all these electrolytes. It also shows that for most electrolytes the main peak position was at ~6.1°. The exception to this were $K^+$ and $Cs^+$ solutions as well as mixtures of other electrolytes with Cs electrolyte. In this case, the peaks were around 7.2°.

## Measurements of ion flux

For permeation measurements, the membranes were affixed over a 0.5 cm diameter hole in a polymethyl methacrylate (PMMA) plate. The plate was then mounted in a polytetrafluoroethylene (PTFE) cell that consisted of two compartments. One compartment (the feed) was filled with 12 mL of salt solution with concentration of 1.0 M for monovalent/divalent cations or 0.1 M for trivalent and tetravalent cations, whereas the other (permeate) contained 12 mL of deionized water. The ion concentration in the permeate solution was monitored periodically using ICP-OES and AAS. Each ion flux experiment was repeated with at least 10 different membranes. The flux, $J$, was calculated using the expression: $J = C V A^{-1} t^{-1}$, where $C$ is the ion concentration in the permeate water, $V$ the volume of the solution in the permeate (12 mL), $A$ the membrane area and $t$ the testing time. From the flux, we estimated the permeability for each ion from the equation[42]: $J = P l^{-1} \Delta C$, where $D$ is the diffusion constant, $\Delta C = C_0 - C_l$ is the difference in concentration

between feed and permeate chambers and $l$ is the membrane thickness. The diffusion constant $D$ is extracted from the expression: $P = D K$, where $K$ is the sorption coefficient, which satisfies $K C^{(m)} = \frac{1}{2} (C_0 + C_l)$, with $C^m$ the average ion concentration in the membrane.

We measured concentration of ions inside the membrane with ICP-OES. We then estimated the ion concentration as a function of channel volume (Supplementary Fig. 11). To that end we converted ion content from millimoles per gram of membrane obtained by ICP-OES in $C$[mmol/g] to millimoles per cubic centimetre of channel volume $C$[mmol/cm³] using: $C[\text{mmol/cm}^3] = C[\text{mmol/g}] \times \rho_{ver} \times d/(d - \theta_{ver})$, where the density of vermiculite is taken as $\rho_{ver} \approx 2.5 g cm^{-3}$, the interlayer spacing $d$ is obtained from XRD, and the monolayer thickness of vermiculite is approximated as $\theta_{ver} \approx 1 nm$.

## Electrochemical impedance measurements

Electrochemical impedance measurements of the membranes were performed following the methods reported in ref. 51. Zr-Ver and Ir-Ver membranes were cut into four equal pieces and then stacked on top of each other to form stacks with thickness from 1 to 4 layers of the original membrane. The stacks were soaked with different electrolyte solutions and then clamped in between two stainless steel plates in a custom-made cell (Supplementary Fig. 12a). Reference experiments were performed using the same methods but using polyethersulfone (PES) membranes instead of vermiculite membranes. Supplementary Fig. 12 shows that the resistance scales linearly with thickness for all membranes, which allows extracting the resistance per layer from the slope of this dependence. We then estimated the diffusion coefficient ($D$) using the expression: $D = LRT(RAz^2F^2C)^{-1}$, where $L$ is the membrane thickness, $R$ is the universal gas constant, $T$ is the absolute temperature, $A$ is the membrane area, $z$ is the charge number of the ions, $F$ is Faraday's constant, and $C$ is the ion concentration within the membrane. This analysis yields $D$ of about $2 \times 10^{-12} m^2 s^{-1}$. These values are consistent with the $D$ estimated from our flux measurements and the independently measured ion concentrations. Reference experiments with PES membranes yielded $D$ about ~100 times larger, consistent with literature values[52].

## Osmotic flow measurements

Water vapour flux was quantified gravimetrically using a $1 cm^2$ Zr-Ver membrane under a nitrogen atmosphere in a glove box. Supplementary Fig. 9 shows that the water permeation is low, approximately ≈100 g m$^{-2}$ h$^{-1}$, confirming that osmotic effects are minimal under the experimental conditions.

## Mixed salt experiments

We measured ion flux through Zr-Ver and Sn-Ver membranes in equimolar (1 M total) mixtures of two different monovalent salts. For all mixtures containing $Cs^+$ or $K^+$, the interlayer spacing collapses (Supplementary Fig. 10). Under these conditions, both membranes yield the same selectivity based on Gibbs hydration energy, as expected given that the channels are now one water layer thick. The only mixture that retains the original interlayer spacing in Sn-Ver is the $Na^+/Li^+$ mixture (Supplementary Fig. 10c, d). In this case, Sn-Ver exhibits a much larger $Li^+$ flux than Zr-Ver, consistent with its permeability in single-salt experiments. However, $Na^+$ permeates approximately twice as fast as $Li^+$ in Sn-Ver. This represents an intermediate behaviour between the selectivity of Sn-Ver in single-salt experiments, where $Li^+$ permeated about six times faster than $Na^+$, and that of Zr-Ver membranes, where $Na^+$ permeated roughly ten times faster than $Li^+$. This difference likely arises from competition for adsorption sites between ions within the confined channels. Less strongly hydrated $Na^+$ ions may preferentially interact with the channel walls, enhancing their local concentration and flux. Similar competitive adsorption effects are well documented in mineral–water interfaces, where the presence of one ion alters the binding and transport of others[53,54].

## Elastic modulus of vermiculite membranes

Membranes with an area of 1 cm² and 2 μm in thickness were glued onto silicon wafers using Stycast 1266. The mechanical properties of the membrane were characterised in liquid environment (DI water) using atomic force microscopy (AFM) in mapping mode. The scanning process included 200 scan lines, with 200 data points per line, and a map scan rate of 0.31 Hz. The AFM experiments were conducted using Bruker Instruments (Multi-Mode 8 SPM), equipped with a Bruker's SCM-PIT-V2 probe whose spring constant was calibrated via the thermal noise method. During imaging, a setpoint force of 100 nN was applied, and the Fast Force Mapping mode was used to measure the mechanical properties of the sample. The indentation depth was determined from the difference between the displacement of the sample and the deformation of the AFM probe, which was measured via a detector in the instrument that enables observing the probe's deformation. These curves were then calibrated to establish the indentation depth's zero point.

Supplementary Fig. 13 shows the representative force curves for different vermiculite membranes. To extract the modulus from the force curves, we modelled the tip as a sphere in the Hertz model. This yields the relation: $F = 4/3 \, E^* \sqrt{R\delta^3}$, where $F$ is the applied force, $\delta$ is the indentation depth, $R$ is the probe radius, and $E^*$ is the effective modulus. The effective modulus accounts for both the sample and probe material properties. However, a posteriori, we find that our membranes, which are rather thick, have moduli of ~0.1 GPa, whereas that of the probe is much stiffer, ~170 GPa. We can therefore neglect the effect of probe on the membranes' moduli. The sample's Young's modulus $E_s$ is then calculated from the formula $E_s = E^*(1-v^2)$, where $v \approx 0.28$ is the Poisson ratio for vermiculite[55,56].

## Data availability

All data supporting the findings of this study are available within the article, the Supplementary Information file or in the database under accession code https://zenodo.org/records/17518885.

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

## Acknowledgements

This work was supported by the National Natural Science Foundation of China (Nos. 22278275, 22022810, and 22494712). Z.L. acknowledges the funding from Sichuan University (2023SCUH0075, SCU2025QNXM-2). Y.C.Z. acknowledges the National Natural Science Foundation of China (No. 12474026), the Natural Science Foundation of Guangdong Province (Basic and Applied Basic Research Foundation 2023A1515011465, 2025A1515011749), the Young Top Talents Program 2021QN02C068. F.C.W. acknowledges the National Natural Science Foundation of China (Nos. 12388101, 12241203) and the Fundamental Research Funds for the Central Universities (WK2090000087). M.L.-H. acknowledges the UKRI (EP/X017745) and The Royal Society (URF\R1\201515). The authors acknowledge Dr. Yunfei, Tian from Analytical and testing centre, SCU for valuable assistance of AFM measurement and analysis. The authors acknowledge the valuable assistance of the ICP-OES and Atomic Absorption Spectroscopy (AAS) Instrumentation Group, specifically Xi Wu, Hao Zeng, Jiahui Yang, and Xu Hou, at the Analytical and Testing Center of Sichuan University.

## Author contributions

Z.L. conceived and initiated the project. M.L.-H., Z.L., and Y.C.Z. directed the project. Y.M.T. and M.C. conducted the experiments. J.H.Q., F.M.P. and F.C.W. provided theory input. Z.L., Y.C.Z., M.L.-H., Y.M.T., G.W.Y., L.-Y.C., and E.H. analysed experimental data. M.L.-H. and Z.L. wrote the manuscript with input from all the authors.

## Competing interests

The authors declare no competing interests.
