## [Transparent Peer Review file · Nature Communications]

Tuneable ion selectivity in vermiculite membranes intercalated with unexchangeable ions

Corresponding Author: Professor Marcelo Lozada-Hidalgo

Version 0:

Reviewer comments:

Reviewer #1

(Remarks to the Author)

In this manuscript, the authors present a study of vermiculite-based membranes intercalated with high-valence, unexchangeable cations, yielding mechanically stable and tunable nanochannels one to two water layers thick with good ion selectivity. The authors demonstrate that ion permeability across these membranes spans five orders of magnitude and correlates strongly with the Gibbs free energy of hydration, offering a route to size- and energy-based ion discrimination. Most strikingly, despite identical interlayer spacing, membranes intercalated with different ions show distinct monovalent ion selectivity sequences, attributed to variations in membrane stiffness and the entropy of hydration of the intercalated ions. This introduces an entropically mediated selectivity mechanism, reminiscent of biological ion channels, that extends beyond traditional charge and size exclusion models. The study offers a conceptual advance by linking hydration entropy, membrane elasticity, and ion transport within nanoscale confinement, providing a new framework for understanding and designing selective membranes. This work is likely to be of interest to the fields of nanofluidics, membrane science, and bioinspired materials, particularly in applications related to water purification, critical ion recovery, and ionic separations. Publication would be recommended, pending revision to address the following issues:

- While the authors argue that the observed selectivity differences arise from differences in ion concentrations within the membrane rather than diffusion coefficients, the latter are not directly measured. The use of time-resolved spectroscopy or electrochemical impedance techniques to extract independent diffusion constants could strengthen this claim.
- The manuscript attributes the ~5% mass loss of unexchangeable ions to surface-bound species, but this assumption is not experimentally verified. Techniques like depth-profiling XPS or STEM-EDS mapping could more rigorously differentiate between surface and interlayer ion distributions, especially given the potential for surface-mediated exchange.
- The introduction omits most of the relevant reports in the literature on ion separation with vermiculite membranes. Monovalent ion separations with vermiculite membranes: *Adv. Mater.* 37 (2025) 2417994. In that work, ions of like charge are effectively separated, counter to the impression from the introduction that this has not been achieved. Also, the first report of stable vermiculite membranes for ion separations can be found here: *ACS Nano* 16 (2022) 18266. Other phyllosilicate membrane ion separations: *ACS Appl. Mater. Interfac.* 15 (2023) 57144.
- The study relies heavily on the interpretation that Gibbs hydration energy dominates selectivity, but does not address possible ion pairing or specific ion-ion interactions within the membrane environment, especially in mixed salt experiments. Including control experiments with non-coordinating counterions (e.g., nitrate instead of chloride) could help rule out pairing effects as a contributor to observed permeability trends.
- Although the authors state that Cl⁻ accompanies cation transport to preserve electroneutrality, a direct analysis of anion-specific permeability (e.g., with differing anions at constant cation) would enrich the understanding of ion transport mechanisms and test the robustness of the hydration-driven selectivity model.

Reviewer #2

(Remarks to the Author)

In this paper, the authors show that Zr^{4+} , Sn^{4+} , Ir^{4+} , and La^{3+} ions, once intercalated into vermiculite membranes, become effectively unexchangeable, thus creating stable channels that can be used to regulate ion transport and selectivity. Moreover, despite having similar interlayer spacing, vermiculite membranes intercalated with different types of intercalated ions show distinct monovalent ion selectivity sequences due to the membranes' stiffness and the hydration entropy of the intercalated ions. The reported results are very interesting and highly important. I would like to recommend the publication of this paper in Nature Communications after a minor revision.

1. The manuscript states that the bright spots correspond to Zr^{4+} ions adsorbed on vermiculite nanosheets (Fig. 1b). However, this identification appears to be based solely on image contrast, without supporting evidence such as EDX to confirm the elemental composition of these regions. Moreover, the characteristic structure of the vermiculite nanosheet can not be distinguished well in the micrograph, making it difficult to verify the presence of the host aluminosilicate layers. More evidence is suggested to be given to support the claim "the ions formed domains with quasi-hexagonal symmetry in the vermiculite layer separated by cation-vacancy regions".
2. In Fig. 1d and Fig. 1e, the authors state that "unexchangeable ions" such as Zr^{4+} remain stably intercalated within the vermiculite nanochannels and can't be replaced by outside ions such as Li^+ , Na^+ , K^+ , Cs^+ , and Sr^{2+} ions. This conclusion is supported by the unchanged XRD patterns of Zr-Ver membranes after 180 days of immersion in 1 M LiCl. However, in Fig. 2a, the XRD spectra show a reduction in interlayer spacing when the membrane is immersed in Cs^+ and K^+ electrolytes. The change is attributed to the so-called 'collapse' of the interlayer space, caused by the easy dehydration of Cs^+ and K^+ , which then adsorb as inner sphere complexes and form polar bonds with the structural oxygen atoms in the vermiculite layers. Similarly, in Fig. 2d, interlayer spacing varies across different salt solutions. These observations appear inconsistent with the earlier claim that Zr^{4+} ions are unexchangeable. If K^+ and Cs^+ cannot replace Zr^{4+} in the interlayer, it is unclear how they could induce a collapse in interlayer spacing. These points should be clarified.
3. The authors mentioned that Sn^{4+} ions are unexchangeable ions in the vermiculite nanochannels and ion exchange process is performed by immersing the Li-Ver membrane in various chloride solutions. However, $SnCl_4$ is highly hydrolytically unstable in water. Upon contact with water, $SnCl_4$ readily hydrolyzes to form insoluble hydroxides or hydrated oxides such as $Sn(OH)_4$ or $SnO_2 \cdot xH_2O$, along with the release of HCl. This raises concerns about whether Sn^{4+} can exist as free ions in aqueous solution long enough to effectively intercalate into the vermiculite layers via ion exchange. More details should be provided about the conditions used to prepare the Sn-Ver membranes and supporting evidence that Sn^{4+} ions were successfully and stably incorporated into the interlayer structure.
4. There are discrepancies between Fig. 1b-e and the corresponding descriptions in the text.
5. The results presented in Fig. 2a-e should be introduced in a logical and sequential order consistent with their labeling. The current text introduces Fig. 2b, 2d, and 2e before 2a and 2c, which may confuse readers. It is recommended to either revise the text or restructure the figure layout to improve clarity and coherence of the paper.

Reviewer #3

(Remarks to the Author)

This study presents the development of vermiculite laminate membranes with tunable interlayer widths through the incorporation of non-exchangeable cations. The fabricated membranes, particularly those intercalated with Zr^{4+} , demonstrated exceptional performance in separating ions of different valences, especially in the presence of Cs^+ . Notably, the authors observed varying trends in the selectivity of monovalent ions depending on the specific cation incorporated. For instance, Li^+ ions permeated most rapidly through membranes containing Ir^{4+} and Sn^{4+} , whereas their transport was the slowest in membranes with Zr^{4+} and La^{3+} . These differences were attributed to variations in membrane stiffness and the hydration entropy of the intercalated cations.

In my opinion, while the observed transport phenomena are potentially of interest to the research community, the mechanistic interpretation remains relatively weak. Therefore, I believe the current version requires substantial revision and is not yet suitable for publication.

1. The reviewer is surprised that the laminate membranes retained their structural integrity during the ion exchange process, for example, during the exchange of Li^+ with Zr^{4+} . Given the dynamic nature of this process, it is possible that membrane delamination occurred. What mechanisms maintain the structural stability of the membranes during ion exchange?
2. The observed change in interlayer spacing in the presence of monovalent ions is intriguing, but the underlying mechanism remains unclear. The authors suggest that dehydrated K^+ or Cs^+ ions form polar bonds with structural oxygen atoms in the vermiculite layers. However, two concerns arise: (1) On Page 3, the manuscript states that all adsorption sites in the vermiculite lattice are occupied by high-valence cations (e.g., Zr^{4+}) for charge neutrality. Then, how can additional monovalent cations (e.g., Cs^+) bond with oxygen atoms in the vermiculite layers? (2) For interlayer spacing to decrease, Cs^+ ions would need to bridge oxygen atoms across adjacent layers. It is unclear whether these interactions are strong enough to draw layers closer, especially in the presence of tightly bound Zr^{4+} ions.
3. According to Fig. 2e, monovalent ion permeation rates appear largely unaffected by channel width. However, the presence of Cs^+ reduces the interlayer spacing from 5.8 Å to 3.2 Å, a substantial change. One would expect a significant decline in permeation under such confinement. Could the authors explain this apparent inconsistency?
4. Extended Data Fig. 7: Rather than expressing ion concentration in the membranes as mmol ions/g membranes, it would be more accurate and informative to report ion concentrations within the nanochannels (ions/channel volume). This is particularly important given that channel widths vary depending on the monovalent ion, which directly impacts ion transport behavior.
5. According to Fig. 2a, the interlayer spacing of Zr-Ver membranes in the presence of KCl is 12.2 Å. However, the discussion at the bottom of Page 5 states that the interlayer spacing of all four membranes is the same under KCl ($d \approx 14.8$ Å). Please clarify which value is correct and reconcile this inconsistency.
6. On Page 6, the authors discuss the potential impact of unexchangeable ion hydration Gibbs energy on ion selectivity, concluding that it does not influence selectivity. However, could the authors elaborate on the mechanisms by which this

property might be expected to affect selectivity?

7. In Fig. 3a (inset), the authors present the Young's modulus of the membranes. It is unclear how this macroscopic mechanical property directly relates to the nanoscale channel elasticity that would influence ion selectivity. Could the authors clarify this connection?

8. Following the previous point, the manuscript introduces a separate concept water stiffness, defined as the energy required to disrupt water's hydrogen-bonding network. The authors suggest that ions with highly ordered hydration shells increase water stiffness and that this is consistent with their observations. However, this work does not appear to include any direct characterization of water stiffness, and the relationship between water stiffness, membrane stiffness, and channel elasticity is not clearly established. The proposed consistency, therefore, appears speculative. Further clarification is needed.

9. The core mechanism proposed is very vague: "On this basis, we attribute the correlation between membrane stiffness and selectivity to entropy-driven effects. These effects likely stem from a complex interplay of factors, including variations in water viscosity and interactions between unexchangeable and permeating ions, all occurring within the nanoscale confinement of the membrane channels." The authors need to clarify the following questions: how does membrane stiffness affect ion selectivity? How does variation in water viscosity impact ion selectivity? How does interactions between unexchangeable and permeating ions affect selectivity? And how does nanoscale confinement amplify these effects? Ideally, these points should be addressed with more quantitative or simulation-based support.

10. The authors should carefully check figure citations throughout the manuscript. For instance, on Page 3, the disappearance of the XRD signal in K-Ver should refer to inset Fig. 1d, not inset Fig. 1c. The retention of the XRD signal in Zr-Ver should be cited as Fig. 1d and Extended Data Fig. 5, not Fig. 1c and Extended Data Fig. 4. The STEM image should be cited as Fig. 1b, not Fig. 1d and e.

Reviewer #4

(Remarks to the Author)

This manuscript reports on the novel finding that certain ions can strongly intercalate with vermiculite, leading to stable membranes with ion selectivity. The stability of the membranes, interlayer spacing, and ion selectivity is characterized, revealing intriguing selectivity trends. In particular, Cs ions are found to decrease spacing leading to large selectivity. Even more intriguingly, La and Zr intercalated vermiculite is found to have much lower permeability to Li compared to other monovalent ions, than Sn and Ir intercalated vermiculite. The difference is traced to the partitioning of Li and correlates with the entropy of hydration of the intercalating ions and the membrane stiffness. This result constitutes clear evidence of effects beyond ion size and charge in ion transport. Overall, the work is thorough and presents intriguing new phenomena related to ion transport in nanoscale channels that is relevant to ion-ion separations. The reviewer recommends publication after addressing the comments below.

The membrane stability is reported in salts of monovalent ions, and in SrCl₂. The claims of stability will be strengthened by including exposure to CaCl₂ and MgCl₂, which are ubiquitous in many practical applications. Furthermore, it will be helpful to address whether the membranes are stable in solutions of cations with +3 or +4 charges, as well as in acidic and basic solutions.

Some of the data relevant to the study are not provided. For example, a consolidated table of interlayer spacings should be added. Similarly, it would be very useful to add a table of permeabilities, along with the ion concentration in the membrane and associated interlayer spacing. Another example, the methods section states that 10 mM or 1 M solutions were used for permeation tests, but it is not clear which concentration is used in which experiment.

Was osmosis of water observed during any of the ion permeation experiments? If yes, it would be very interesting to report the measured water fluxes. Was stirring employed to ensure good mixing, or was it not found to be necessary?

Was the permeability observed to be independent of the membrane thickness? If not, it would suggest percolation effects, defects, or variation in the membrane structure.

The manuscript largely reports ion permeation experiments with a single cation at a time. To ascertain selectivity, it is necessary to perform mixed ion experiments to assess whether the selectivity observed in single salt experiments also carries over to mixed salt experiments.

Was the effect of salt concentration and pH on permeation investigated? In particular, measuring the partitioning of ions, interlayer spacing, and permeability, along with estimating the diffusivity as a function of salt concentration will reveal whether the salt concentration affects the interlayer spacing and ion transport, what kind of adsorption isotherms describe ion uptake into intercalated vermiculite, and whether transport in the dilute limit can explain transport at higher salt concentrations.

The hypothesis of entropic effects in the hydration of the intercalating ion on the differences in Li ion transport is very intriguing. However, the reason for the behavior is not pursued apart from correlation to entropy of hydration and mechanical stiffness of the membrane. Does the interlayer spacing change in the presence of Li ions, especially in the case of Sn and Ir intercalation where the membranes are more flexible? Do the observations correlate with the thermodynamic behavior of mixed salt solutions of the intercalating ion and lithium indicating that it may be a bulk effect, or is nanoconfinement essential? What explains the order of magnitude difference in ion partitioning for Sn and Ir intercalated membranes – can it be attributed to interlayer spacing, or are other effects essential? Similarly, does Li partition more favorably than the other monovalent ions in the Sn and Ir intercalated membranes and why (and if not, it must diffuse faster given the data in Fig. 3)? Similarly, temperature dependence to extract the free energy change of partitioning of ions will likely provide additional

insights.

Version 1:

Reviewer comments:

Reviewer #1

(Remarks to the Author)

The authors have satisfactorily addressed the issues raised during the initial review. This work is now suitable for publication.

Reviewer #2

(Remarks to the Author)

The authors have addressed my comments properly, and I would like to recommend the publication of this manuscript in Nature Communications.

Reviewer #3

(Remarks to the Author)

The authors have addressed most of my concerns. I have two minor comments that I would like the authors to further clarify before this work is published:

1. Monovalent ion permeation with varying channel widths

Referring to my previous Comment 3, I remain surprised by the relatively unaffected permeation of monovalent ions with different channel widths (Fig. 2e). I understand that ion permeation is governed by the thermodynamic partitioning step, which depends on ion dehydration and ion–pore interactions. A smaller pore is typically expected to impose a higher dehydration penalty. At the same time, ions with fewer surrounding water molecules should interact more strongly with the pore walls, partially compensating for this penalty. The authors suggest that the observed unaffected monovalent permeation arises from a balance: the additional dehydration penalty is approximately offset by stronger ion–pore interactions, resulting in a similar overall free energy change for both channel sizes. My question is: is this balance merely coincidental in this case, or does it represent a more universal principle?

2. Relation between elasticity and selectivity

Referring to my previous Comment 9, I appreciate the improved discussion of the mechanism, but I still find the link between membrane elasticity and ion selectivity insufficiently explicit. The current revision convincingly shows that pore elasticity influences ion selectivity, and that incorporating different non-exchangeable ions alters the elasticity of the membrane. However, it remains unclear, at least to me, and potentially to readers, why Li^+ permeates most slowly through membranes with higher Young's modulus (Zr-Ver and La-Ver) and most rapidly through those with lower Young's modulus (Sn-Ver and Ir-Ver). A clearer mechanistic explanation of this trend would strengthen the interpretation.

Reviewer #4

(Remarks to the Author)

The authors have addressed all the reviewer comments satisfactorily and the reviewer recommends publication without further revision.

Version 2:

Reviewer comments:

Reviewer #3

(Remarks to the Author)

The authors have addressed all my comments, and I recommend this manuscript for publication.

Response to comments from Reviewer #1

In this manuscript, the authors present a study of vermiculite-based membranes intercalated with high-valence, unexchangeable cations, yielding mechanically stable and tunable nanochannels one to two water layers thick with good ion selectivity. The authors demonstrate that ion permeability across these membranes spans five orders of magnitude and correlates strongly with the Gibbs free energy of hydration, offering a route to size- and energy-based ion discrimination. Most strikingly, despite identical interlayer spacing, membranes intercalated with different ions show distinct monovalent ion selectivity sequences, attributed to variations in membrane stiffness and the entropy of hydration of the intercalated ions. This introduces an entropically mediated selectivity mechanism, reminiscent of biological ion channels, that extends beyond traditional charge and size exclusion models. The study offers a conceptual advance by linking hydration entropy, membrane elasticity, and ion transport within nanoscale confinement, providing a new framework for understanding and designing selective membranes. This work is likely to be of interest to the fields of nanofluidics, membrane science, and bioinspired materials, particularly in applications related to water purification, critical ion recovery, and ionic separations. Publication would be recommended, pending revision to address the following issues:

We thank the reviewer for this positive assessment of our work and for the focused and constructive critique of the manuscript. We have done our best to address the points raised.

- While the authors argue that the observed selectivity differences arise from differences in ion concentrations within the membrane rather than diffusion coefficients, the latter are not directly measured. The use of time-resolved spectroscopy or electrochemical impedance techniques to extract independent diffusion constants could strengthen this claim.

Figure R1.1. (a) Experimental setup. (b) Resistance of typical membrane stack (Zr-Ver) soaked in electrolyte solution (LiCl) as a function of number of stacked membranes. The membrane resistance was extracted from the slope of the linear fit of the resistance vs. the numbers of membranes stacked layer-by-layer. Inset, Nyquist plot of membrane stacks from which resistance in main panel is extracted. (c) Diffusion coefficients for different ions in Zr-Ver, Ir-Ver and PES-membrane (bulk) membranes extracted from resistance data.

Following the reviewer's suggestion, we have performed electrochemical impedance spectroscopy measurements of monovalent ion permeation in Zr-Ver and Ir-Ver, and for reference, also in polyethersulfone (PES) membranes. To obtain accurate measurements of the samples, we followed the method described in Dai, Q. *et al. Nat. Comms.* 2020, 11, 13. In brief, we cut the membranes into four equal pieces that are stacked on top of each other to form stacks with thickness from 1 to 4 layers of the original membrane. The stacks are soaked in the

electrolyte solution and clamped in between two stainless steel plates. Fig. R1-1 shows that the membrane resistance scales linearly with thickness for all membranes, which allows extracting the resistance per layer of membrane from the slope of this dependence. We then estimate the diffusion coefficient (D) using the expression: $D = LRT(R_{\Omega}Az^2F^2C)^{-1}$, where L is the membrane thickness, R is the universal gas constant, T is the absolute temperature, A is the membrane area, z is the charge number of ions, F is Faraday's constant, and C is the ion concentration within the membrane.

This analysis yields D of about $2 \times 10^{-12} \text{ m}^2/\text{s}$, for the different monovalent ions in our membranes, consistent with the D estimated from our flux measurements and the independently measured ion concentration inside the membrane. This confirms that the observed differences in ion flux are fully explained by concentration differences within the membranes, rather than differences in diffusion coefficients. For the reference samples, we found about ~ 100 times larger D , consistent with literature values (Ref. 48). We have included these data as Supplementary Fig. 12 and this discussion in page 6 in the revised manuscript.

- The manuscript attributes the $\sim 5\%$ mass loss of unexchangeable ions to surface-bound species, but this assumption is not experimentally verified. Techniques like depth-profiling XPS or STEM-EDS mapping could more rigorously differentiate between surface and interlayer ion distributions, especially given the potential for surface-mediated exchange.

To address this comment, we examined the depth profile of unexchangeable Zr ions using XPS (Fig. R1-2). No significant change in Zr concentration was observed after immersion in different electrolytes. With hindsight, this was expected, as Zr constitutes only $\sim 1\%$ of the membrane

mass, and detecting a 5% change of such a small fraction exceeds the technique's sensitivity. The lack of a measurable change in the Zr signal after electrolyte treatment indicates that Zr⁴⁺ loss is indeed minimal, confirming that the ions are effectively unexchangeable.

Following the Reviewer comment, we have removed the statement about the origin of the loss of the ions.

Figure R1-2. a, The XPS spectrum of the Zr-Ver membrane with different depth after different solution immersion. **b,** Zoom in of

area in dashed box in panel a.

- The introduction omits most of the relevant reports in the literature on ion separation with vermiculite membranes. Monovalent ion separations with vermiculite membranes: *Adv. Mater.* 37 (2025) 2417994. In that work, ions of like charge are effectively separated, counter to the impression from the introduction that this has not been achieved. Also, the first report of stable vermiculite membranes for ion separations can be found here: *ACS Nano* 16 (2022) 18266. Other phyllosilicate membrane ion separations: *ACS Appl. Mater. Interfac.* 15 (2023) 57144.

These are indeed very relevant references. We now discuss them in the introduction of the revised manuscript (page 2) as Refs. 32-34. Apologies for the oversight.

- The study relies heavily on the interpretation that Gibbs hydration energy dominates selectivity, but does not address possible ion pairing or specific ion-ion interactions within the membrane environment, especially in mixed salt experiments. Including control experiments with non-coordinating counterions (e.g., nitrate instead of chloride) could help rule out pairing effects as a contributor to observed permeability trends.

Following the Reviewer comment we fixed NO₃⁻ as the counterion and measured permeation of various cations. Fig. R1-3 shows that the selectivity sequence remained the same, with only some cations displaying flux differences of a factor ~ 2 between different salts. These results indicate that ion pairing does not significantly influence membrane selectivity. The data are included as Supplementary Fig. 7 in the revised manuscript.

Figure R1-3. Nitrate salt ion flux of the Zr-Ver membrane

- Although the authors state that Cl⁻ accompanies cation transport to preserve electroneutrality, a direct analysis of anion-specific permeability (e.g., with differing anions at constant cation) would enrich the understanding of ion transport mechanisms and test the robustness of the hydration-driven selectivity model.

Following the Reviewer comment, we investigated anion permeability with Li⁺ as the fixed cation (Figure R1-4). Chloride and nitrate salts give similar results, whereas SO₄²⁻, being more strongly hydrated, permeates more slowly than Cl⁻, and Li⁺ transport is correspondingly reduced to preserve electroneutrality. These data have been added as Supplementary Fig. 8b in the revised manuscript.

Figure R1-4. Flux of lithium salts with different anions through Zr-Ver membranes.

Response to comments from Reviewer #2

In this paper, the authors show that Zr^{4+} , Sn^{4+} , Ir^{4+} , and La^{3+} ions, once intercalated into vermiculite membranes, become effectively unexchangeable, thus creating stable channels that can be used to regulate ion transport and selectivity. Moreover, despite having similar interlayer spacing, vermiculite membranes intercalated with different types of intercalated ions show distinct monovalent ion selectivity sequences due to the membranes' stiffness and the hydration entropy of the intercalated ions. The reported results are very interesting and highly important. I would like to recommend the publication of this paper in Nature Communications after a minor revision.

We are very grateful to the Reviewer for this encouraging assessment of our work. We have carefully addressed all the comments.

1. The manuscript states that the bright spots correspond to Zr^{4+} ions adsorbed on vermiculite nanosheets (Fig. 1b). However, this identification appears to be based solely on image contrast, without supporting evidence such as EDX to confirm the elemental composition of these regions. Moreover, the characteristic structure of the vermiculite nanosheet can not be distinguished well in the micrograph, making it difficult to verify the presence of the host aluminosilicate layers. More evidence is suggested to given to support the claim “the ions formed domains with quasi-hexagonal symmetry in the vermiculite layer separated by cation-vacancy regions”.

Following the Reviewer comment, we have now included EDX measurements (Figure R2-1), which confirm the presence of Zr in the regions corresponding to the bright spots in the image. All the peaks, except Zr (and Cu from the holder), arise from native vermiculite, which contains O ($Z_O=8$), Mg ($Z_{Mg}=12$), Al ($Z_{Al}=13$), Si ($Z_{Si}=14$), with trace elements of Ca ($Z_{Ca}=20$), Ti ($Z_{Ti}=22$), Cr ($Z_{Cr}=24$), Fe ($Z_{Fe}=26$). Moreover, no bright spot can be observed when imaging pristine vermiculite under ADF mode (collection angle 40-200 mrad). Since Zr ($Z_{Zr}=40$) is the only heavy element in the sample, the bright atom columns can be unambiguously attributed to Zr. We thank the Reviewer for pointing this out to us, we agree that this is an important characterisation. We have added this to the revised Supplementary Fig. 1.

Figure R2-1. Extended TEM data taken from the sample shown in Fig. 1 in the manuscript. (a) ADF-STEM image showing the over-view image of Zr-Ver nanosheets on TEM grids. (b) The selected-area electron diffraction taken from the sample. (c) Corresponding EDS spectrum, where the Cu peaks are from the TEM holder and the supporting grids.

Regarding the imaging of the aluminosilicate layers, we found that these cannot be resolved in plan-view images due to their beam sensitivity and complex $[001^*]$ projection. A low electron dose allows clear imaging of the heavy Zr columns without damaging the crystal, but imaging the aluminosilicate layers requires a higher dose. This causes rapid damage and therefore

distinguishing the aluminosilicate layer is not possible. In this work, we applied a low electron dose to image Zr atoms and SAED collected after imaging (Fig. R2-1b) confirms that the aluminosilicate layers preserve a high crystal quality, indicating that our imaging reflects the real arrangement of Zr atoms.

2. In Fig. 1d and Fig. 1e, the authors state that "unexchangeable ions" such as Zr^{4+} remain stably intercalated within the vermiculite nanochannels and can't be replaced by outside ions such as Li^+ , Na^+ , K^+ , Cs^+ , and Sr^{2+} ions. This conclusion is supported by the unchanged XRD patterns of Zr-Ver membranes after 180 days of immersion in 1 M LiCl. However, in Fig. 2a, the XRD spectra show a reduction in interlayer spacing when the membrane is immersed in Cs^+ and K^+ electrolytes. The change is attributed to the so-called 'collapse' of the interlayer space, caused by the easy dehydration of Cs^+ and K^+ , which then adsorb as inner sphere complexes and form polar bonds with the structural oxygen atoms in the vermiculite layers. Similarly, in Fig. 2d, interlayer spacing varies across different salt solutions. These observations appear inconsistent with the earlier claim that Zr^{4+} ions are unexchangeable. If K^+ and Cs^+ cannot replace Zr^{4+} in the interlayer, it is unclear how they could induce a collapse in interlayer spacing. These points should be clarified.

We clarify that K^+ and Cs^+ ions do not replace Zr^{4+} in the interlayer during permeation experiments. The concentration of Zr^{4+} is relatively low because, due to its +4 valence, each Zr^{4+} ion compensates multiple exchange sites in the vermiculite (each with valence -1). As a result, a large number of cation vacancies remain even after Zr^{4+} intercalation, leaving many sites available for additional adsorption—more than those occupied by Zr^{4+} ions. This can be appreciated more clearly in zoom-out STEM images of Zr-Ver membranes (Fig. R2-2).

Fig. R2-2. STEM image of Zr-Ver over larger areas. Bright spots, Zr-atoms. Scale bar, 1 nm.

The observed reduction in interlayer spacing in Cs^+ and K^+ electrolytes is therefore not due to ion exchange with Zr^{4+} , but rather to the adsorption of Cs^+ and K^+ at these vacant sites (see also Fig. R3-1, page 8 below). During interlayer collapse, the areas of the flake covered with Zr^{4+} retain a fixed ≈ 14.8 Å interlayer distance, whereas those with adsorbed Cs^+ , collapse. Since Zr^{4+} -free areas occupy a larger surface area, their collapse governs the overall XRD response. This can be observed in the XRD spectra in Fig. 2c,d and Supplementary Fig. 4. The XRD spectra yield a peak centred at ≈ 12.2 Å, however, the distribution is broader due to larger interlayer distance in the areas with adsorbed Zr^{4+} . Following the Reviewer comment, we have clarified this point in the manuscript on page 5, last sentences in the first paragraph.

3. The authors mentioned that Sn^{4+} ions are unexchangeable ions in the vermiculite nanochannels and ion exchange process is performed by immersing the Li-Ver membrane in various chloride solutions. However, $SnCl_4$ is highly hydrolytically unstable in water. Upon contact with water, $SnCl_4$ readily hydrolyzes to form insoluble hydroxides or hydrated oxides such as $Sn(OH)_4$ or $SnO_2 \cdot xH_2O$, along with the release of HCl. This raises concerns about whether Sn^{4+} can exist as free ions in aqueous solution long enough to effectively intercalate into the vermiculite layers via ion exchange. More details should be provided about the conditions used to prepare the Sn-Ver membranes and supporting evidence that Sn^{4+} ions were successfully and stably incorporated into the interlayer structure.

SnCl_4 can indeed be hydrolytically unstable in water and can, under certain conditions, form $\text{Sn}(\text{OH})_4$ or hydrated SnO_2 . However, we find that our carefully dispersed 1 M SnCl_4 aqueous solutions do not display obvious precipitation. As shown in Figure R2-3, our 1 M SnCl_4 solution remains visibly clear after half a year (Fig. R2-3a), indicating that hydrolysis is either suppressed or proceeded very slowly. Moreover, both XRD (Fig. R2-3b) and TEM (Fig. R2-3c) of the Sn-Ver membranes show no evidence of $\text{Sn}(\text{OH})_4$ or $\text{SnO}_2 \cdot x\text{H}_2\text{O}$ phases. Our TEM images show that the Sn^{4+} ions are incorporated into the crystals as single ions, which can be seen as bright spots from experimental atomic-resolution ADF-STEM images (see Fig. R2-3c below). This discussion has been added in page 8, paragraph 4 in the revised manuscript.

Figure R2-3 (a) Digital photo of a 1 M SnCl_4 aqueous solution taken 6 months after its preparation, (b) XRD pattern of Sn-Ver membrane (blue spectrum) against pure vermiculite PDF (black lines below), (c) ADF-STEM image showing well-dispersed Sn ions as those bright spots from Sn-Ver membranes.

4. There are discrepancies between Fig. 1b-e and the corresponding descriptions in the text.

We have corrected these discrepancies in the text. Much appreciated.

5. The results presented in Fig. 2a-e should be introduced in a logical and sequential order consistent with their labeling. The current text introduces Fig. 2b, 2d, and 2e before 2a and 2c, which may confuse readers. It is recommended to either revise the text or restructure the figure layout to improve clarity and coherence of the paper.

We agree with this suggestion. Fig. 2 has been reordered to follow the logical sequence of the draft. Much appreciated.

Response to comments from Reviewer #3

This study presents the development of vermiculite laminate membranes with tunable interlayer widths through the incorporation of non-exchangeable cations. The fabricated membranes, particularly those intercalated with Zr^{4+} , demonstrated exceptional performance in separating ions of different valences, especially in the presence of Cs^+ . Notably, the authors observed varying trends in the selectivity of monovalent ions depending on the specific cation incorporated. For instance, Li^+ ions permeated most rapidly through membranes containing Ir^{4+} and Sn^{4+} , whereas their transport was the slowest in membranes with Zr^{4+} and La^{3+} . These differences were attributed to variations in membrane stiffness and the hydration entropy of the intercalated cations.

In my opinion, while the observed transport phenomena are potentially of interest to the research community, the mechanistic interpretation remains relatively weak. Therefore, I believe the current version requires substantial revision and is not yet suitable for publication.

Following the Reviewer comment, we have revised the manuscript to strengthen the discussion of the transport mechanism. We believe the updated version provides a clearer interpretation.

1. The reviewer is surprised that the laminate membranes retained their structural integrity during the ion exchange process, for example, during the exchange of Li^+ with Zr^{4+} . Given the dynamic nature of this process, it is possible that membrane delamination occurred. What mechanisms maintain the structural stability of the membranes during ion exchange?

We agree that it is surprising that the membranes retained their structural integrity during ion exchange. However, we are confident about this result. Regarding the mechanism, we studied ion exchange in 2D clays in detail, including STEM images during intermediate stages of the exchange, in our previous work Y.-C. Zou *et al.*, Nat. Mater. 2021, 20, 1677-1682 (Ref. 28). The stability arises for two reasons. First, ion exchange occurs gradually within the interlayer galleries, without disrupting the layered structure, which is bound both by short range electrostatic forces and long-range van der Waals interactions. Second, ion exchange in 2D vermiculite is orders of magnitude faster than in bulk crystals, avoiding the accumulation of local strain that could otherwise delaminate the membrane. This fast but gentle dynamics allows cations to be replaced efficiently while preserving the structural integrity of the laminate.

2. The observed change in interlayer spacing in the presence of monovalent ions is intriguing, but the underlying mechanism remains unclear. The authors suggest that dehydrated K^+ or Cs^+ ions form polar bonds with structural oxygen atoms in the vermiculite layers. However, two concerns arise: (1) On Page 3, the manuscript states that all adsorption sites in the vermiculite lattice are occupied by high-valence cations (e.g., Zr^{4+}) for charge neutrality. Then, how can additional monovalent cations (e.g., Cs^+) bond with oxygen atoms in the vermiculite layers? (2) For interlayer spacing to decrease, Cs^+ ions would need to bridge oxygen atoms across adjacent layers. It is unclear whether these interactions are strong enough to draw layers closer, especially in the presence of tightly bound Zr^{4+} ions.

(1) Following the comment from the Reviewer and also from Reviewer #2, we now see that this point was not clear. The manuscript states the opposite. It is not possible for all adsorption sites to be occupied by Zr^{4+} because the ion's 4+ valence compensates the charge of multiple lattice sites (each site has a -1 valence). As a result, many more sites remain empty than occupied by Zr^{4+} (e.g. Fig. R2-2, page 5 above). Cs^+ ions adsorb on these empty sites.

(2) The reduction in interlayer spacing arises from the dehydration of Cs^+ ions, a well-established phenomenon in clays (refs. 31, 38–41 in the manuscript). Because water molecules are weakly

bound to Cs^+ , the electrostatic attraction between the negatively charged vermiculite layers and the ions strips part of the hydration shell, leading to a collapse of the interlayer space. In our membranes, regions with fixed Zr^{4+} retain their spacing, while Cs^+ -exposed areas collapse, as illustrated schematically in Fig. R3-1. We can see this in the XRD spectra of these samples (e.g. Fig. 2c and Supplementary Fig. 4). The main peak in the presence of Cs^+ is centred around 12.2 Å, but it is broader than for ions that do not induce layer collapse, indicating less uniform thickness, as expected.

Following the Reviewer comment, we now see that these two important points were not clear in the paper. We have clarified them in the revised manuscript in page 3 last paragraph and page 5 first paragraph.

Figure R3-1. Schematic diagram of Cs^+ -induced collapse of the interlayer space of Zr-Ver membrane.

3. According to Fig. 2e, monovalent ion permeation rates appear largely unaffected by channel width. However, the presence of Cs^+ reduces the interlayer spacing from 5.8 Å to 3.2 Å, a substantial change. One would expect a significant decline in permeation under such confinement. Could the authors explain this apparent inconsistency?

This observation may appear inconsistent with expectations based on steric exclusion. However, in charged, hydrophilic systems like vermiculite membranes, ion permeability is not governed solely by steric exclusion but by the thermodynamics of ion partitioning—that is, by the equilibrium concentration of ions inside the membrane. This is shown in Supplementary Fig. 11 and the new Supplementary Fig. 12, which show that the concentration of ions inside the channels, rather their diffusion, determine the permeability of the membrane. In turn, this equilibrium concentration is determined by the difference in the ions' Gibbs energy between the external solution and the membrane (ΔG).

For monovalent ions, which are relatively weakly hydrated compared to higher valence ions, the electrostatic attraction to the negatively charged clay surfaces can be stronger than to water molecules, as discussed above. Cs^+ provides the clearest example. Its strong affinity for the walls drives interlayer collapse and the same attraction results in a high concentration of Cs^+ within the channel, leading to high permeability. Hence, channels of different width can lead to similar permeability if ΔG is similar. This has now been explained in paragraph 2, page 5 of the main text.

4. Extended Data Fig. 7: Rather than expressing ion concentration in the membranes as mmol ions/ g membranes, it would be more accurate and informative to report ion concentrations within the nanochannels (ions/ channel volume). This is particularly important given that channel widths vary depending on the monovalent ion, which directly impacts ion transport behavior.

Fig. R3-2. Estimation of ion concentration inside the channels.

The conversion from ion content expressed as millimoles per gram of membrane ($C_{\text{mmol/g}}$) to millimoles per cubic centimetre of channel volume ($C_{\text{mmol/cm}^3}$) was calculated

using the formula: $C_{\text{mmol/cm}^3} = C_{\text{mmol/g}} \times \rho_{\text{ver}} \times d / (d - \theta_{\text{ver}})$. The density of vermiculite is taken as $\rho_{\text{ver}} = 2.5 \text{ g/cm}^3$, the spacing height d with different ions is taken from the XRD results. The thickness of the vermiculate layer θ_{ver} of the monolayer vermiculite is estimated as $\approx 1 \text{ nm}$. Fig. R3-2 shows these estimates. The concentrations range from 1-100 mmol cm^{-3} for the different membranes.

5. According to Fig. 2a, the interlayer spacing of Zr-Ver membranes in the presence of KCl is 12.2 Å. However, the discussion at the bottom of Page 5 states that the interlayer spacing of all four membranes is the same under KCl ($d \approx 14.8 \text{ Å}$). Please clarify which value is correct and reconcile this inconsistency.

It should say 12.2 Å for KCl. Much appreciated.

6. On Page 6, the authors discuss the potential impact of unexchangeable ion hydration Gibbs energy on ion selectivity, concluding that it does not influence selectivity. However, could the authors elaborate on the mechanisms by which this property might be expected to affect selectivity?

That sentence in the original manuscript (page 6) aimed to emphasise that conventional descriptors fail to explain our results. To avoid confusion, we have removed the reference to hydration Gibbs energy from this paragraph.

7. In Fig. 3a (inset), the authors present the Young's modulus of the membranes. It is unclear how this macroscopic mechanical property directly relates to the nanoscale channel elasticity that would influence ion selectivity. Could the authors clarify this connection?

The bending modulus of a clay layer in the direction i , κ_i , is related to the Young's modulus in the i direction, E_i , by the equation: $\kappa_i = E_i h^3 / (12(1-\nu^2))$, with h the clay thickness and ν its Poisson ratio (e.g., J. L. Suter, et al. J. Phys. Chem. C 2007, 111, 8249-8259). We explain how channel elasticity is related to selectivity in point #9.

8. Following the previous point, the manuscript introduces a separate concept water stiffness, defined as the energy required to disrupt water's hydrogen-bonding network. The authors suggest that ions with highly ordered hydration shells increase water stiffness and that this is consistent with their observations. However, this work does not appear to include any direct characterization of water stiffness, and the relationship between water stiffness, membrane stiffness, and channel elasticity is not clearly established. The proposed consistency, therefore, appears speculative. Further clarification is needed.

Measuring water stiffness, even in bulk water, requires specialised techniques, and extending them to our system would constitute a major independent study. Nonetheless, we argue that using this concept is both justified and informative, based on established thermodynamic descriptors for water and prior experiments of water confined between clay layers.

Water exhibits an unusually high resistance to local deformation due to its extended hydrogen-bond network. This resistance can be quantified by several thermodynamic parameters, including cohesive energy density, isothermal compressibility, heat capacity density, and orientational correlation of its dipoles. Collectively, these quantities capture the energetic cost of disrupting its structured environment or "stiffness". Marcus and others have shown that ions with high solvation entropy form highly ordered hydration shells and induce long-range structuring of surrounding water, increasing water's stiffness (Ref. 46).

Under confinement, and specifically between clay layers, water becomes even more structured. Direct imaging studies using high-resolution AFM (Fukuma *et al.*, *Phys. Rev. Lett.* 2010, 104, 016101) have demonstrated clear layering and orientational order of water at clay surfaces.

Moreover, classic surface force measurements (Israelachvili, *Intermolecular and Surface Forces*, Academic Press 2011, Ch. 15) show that confined water strongly resists compression over nanometre-scale distances with forces that exceed expectations from bulk. Hence, these highly ordered water layers are expected to display even more pronounced stiffness changes than bulk water when ions with highly ordered solvation shells are introduced.

Although we do not directly probe water structure in our membranes due to the complexity of such measurements, our data show that membranes intercalated with ions forming strongly ordered hydration shells consistently display higher Young's moduli. This supports the view that structured interlayer water contributes directly to membrane stiffness and that ions solvated in these water layers significantly affect their stiffness. We therefore propose that the mechanical response of our membranes reflects both the intrinsic elasticity of the clay layers and, crucially, the resistance to deformation provided by the structured water phase. As water stiffness increases with the hydration entropy of the intercalated ions, these thermodynamic properties offer a consistent framework for interpreting the observed mechanical trends.

9. The core mechanism proposed is very vague: "On this basis, we attribute the correlation between membrane stiffness and selectivity to entropy-driven effects. These effects likely stem from a complex interplay of factors, including variations in water viscosity and interactions between unexchangeable and permeating ions, all occurring within the nanoscale confinement of the membrane channels." The authors need to clarify the following questions: how does membrane stiffness affect ion selectivity? How does variation in water viscosity impact ion selectivity? How do interactions between unexchangeable and permeating ions affect selectivity? And how does nanoscale confinement amplify these effects? Ideally, these points should be addressed with more quantitative or simulation-based support.

We agree with the Reviewer that the mechanism could be clearer and have done our best to clarify the proposed mechanism in the revised manuscript. Specifically, regarding the questions from the Reviewer.

Stiffness vs selectivity. The link between membrane stiffness and ion selectivity was first observed in biological porins, where channels of similar size exhibit different monovalent ion selectivity sequences. This behaviour is captured by analytical models that describe the system's Hamiltonian with three key components: ion hydration free energy, ion-channel electrostatic interaction, and channel elasticity (A. Laio & V. Torre, *Biophysical Journal* 1999, 76, 129-148). Elasticity is critical because thermal fluctuations cause the channel radius to fluctuate around its equilibrium value, which changes the ion's Gibbs energy as: $H_e(x) = (1/2) k(x) [R(x) - R_0(x)]^2$, where $k(x)$ is the stiffness as a function of position in the channel, x , and $R_0(x)$ is the equilibrium radius. All else being equal, this can lead to different monovalent ion selectivity sequences, as shown in Fig. 4&5 in Laio et al (Ref. 43).

Analogous effects have been reported in clay systems. For example, M. L. Whittaker, et al. (PNAS 2019, 116, 22052-22057) reported time-resolved X-ray scattering experiments that showed that ion exchange in clays is coupled to the dynamic fluctuation of the average basal plane spacing. This has been modelled, for example using simulation boxes with over 10^5 atoms and timescales of the order of hundreds of picoseconds (Suter et al. *J. Phys. Chem. C*, 2007, 111, 8249-8259). To model our system, in addition to this, we would need to incorporate the many different ions (permeating and unexchangeable) and let the sheet relax. We believe that a comprehensive theoretical treatment of this system warrants a dedicated follow up study.

Water stiffness vs selectivity and confinement. Water stiffness affects Young's modulus (point #8), which affects channel elasticity, and thus selectivity, as described above.

Interactions between unexchangeable-permeating ions vs selectivity. The activity coefficient of permeating ions should be affected by the presence of highly charged unexchangeable ions, as predicted by Debye-Huckel theory. However, this effect is not central to our findings, since membranes' properties differ even when intercalated with ions of the same charge (e.g. Zr-Ver and Ir-Ver). To avoid confusion, we have removed this statement from the manuscript.

Confinement. Water becomes more ordered under confinement. Entropic effects from ions with large hydration entropy should be more evident in these ordered films, as discussed above. Following the Reviewer comment, we have extended the discussion of the mechanism in the revised manuscript, page 7.

10. The authors should carefully check figure citations throughout the manuscript. For instance, on Page 3, the disappearance of the XRD signal in K-Ver should refer to inset Fig. 1d, not inset Fig. 1c. The retention of the XRD signal in Zr-Ver should be cited as Fig. 1d and Extended Data Fig. 5, not Fig. 1c and Extended Data Fig. 4. The STEM image should be cited as Fig. 1b, not Fig. 1d and e.

We thank the Reviewer for noticing this. We have carefully checked and corrected all figure references throughout the manuscript.

Response to comments from Reviewer #4

This manuscript reports on the novel finding that certain ions can strongly intercalate with vermiculite, leading to stable membranes with ion selectivity. The stability of the membranes, interlayer spacing, and ion selectivity is characterized, revealing intriguing selectivity trends. In particular, Cs ions are found to decrease spacing leading to large selectivity. Even more intriguingly, La and Zr intercalated vermiculite is found to have much lower permeability to Li compared to other monovalent ions, than Sn and Ir intercalated vermiculite. The difference is traced to the partitioning of Li and correlates with the entropy of hydration of the intercalating ions and the membrane stiffness. This result constitutes clear evidence of effects beyond ion size and charge in ion transport. Overall, the work is thorough and presents intriguing new phenomena related to ion transport in nanoscale channels that is relevant to ion-ion separations. The reviewer recommends publication after addressing the comments below.

We sincerely thank the Reviewer for their encouraging assessment of our work. We have carefully addressed all the comments and have revised the manuscript accordingly. Detailed responses to each point are provided below.

The membrane stability is reported in salts of monovalent ions, and in SrCl₂. The claims of stability will be strengthened by including exposure to CaCl₂ and MgCl₂, which are ubiquitous in many practical applications. Furthermore, it will be helpful to address whether the membranes are stable in solutions of cations with +3 or +4 charges, as well as in acidic and basic solutions.

Following the Reviewer's comment, we investigated the stability of Zr-Ver membranes upon

exposure to 1 M CaCl₂, 1 M MgCl₂, 1 M LaCl₃, 0.1 M IrCl₄, as well as acidic (HCl, pH ≈ 3) and basic (NaOH, pH ≈ 12) solutions. We quantified Zr⁴⁺ loss using mass spectrometry and found it to be below ~5% in all cases (Fig. R4-1, now revised Fig. 1e). We also monitored the membranes visually over 30 days and confirmed that they remained intact, with no signs of swelling, delamination.

Figure R4-1. Zr⁴⁺ mass loss after immersing the ionic solutions for 24h.

Some of the data relevant to the study are not provided. For example, a consolidated table of interlayer spacings should be added. Similarly, it would be very useful to add a table of permeabilities, along with the ion concentration in the membrane and associated interlayer spacing. Another example, the methods section states that 10 mM or 1 M solutions were used for permeation tests, but it is not clear which concentration is used in which experiment. Following the Reviewer comment, we have added a table (See Supplementary Table 1) summarising ion permeabilities, membrane ion concentrations, and associated interlayer spacings.

The statement that some experiments used 10 mM solutions was an error. Experiments were performed with 1M solutions in all cases, except for salts with +3 and +4 valence cations, in which we used 0.1 M to avoid making the solutions acidic—although with hindsight, we now know it doesn't matter (see Fig. R4-6 below). We have clarified this in the Methods section.

Was osmosis of water observed during any of the ion permeation experiments? If yes, it would be

very interesting to report the measured water fluxes. Was stirring employed to ensure good mixing, or was it not found to be necessary?

We did explore this possibility, but did not observe osmotic flux, most likely because it was too

small. Following the Reviewer comment, we now tested water vapour flux gravimetrically, using a 1 cm² Zr-Ver membrane. Fig. R4-2 shows that the water permeation was low, at about 100 g m⁻² h⁻¹. Following the Reviewer comment we have included this figure as Supplementary Fig. 9 in the revised manuscript. Regarding the second question, we did test both with and without stirring and did not observe any significant differences.

Figure R4-2. Gravimetry measurements of water vapour permeation through Zr-Ver membranes.

Was the permeability observed to be independent of the membrane thickness? If not, it would suggest percolation effects, defects, or variation in the membrane structure.

The ion permeability scales linearly with membrane thickness in the range of 0.5–2 μm, indicating that transport is governed by through-plane diffusion and not dominated by percolation effects, defects, or structural variations (Fig. R4-3). We have included this important point as Supplementary Fig. 6 in the revised manuscript.

Figure R4-3. the ion flux dependent on the thickness of Zr-Ver membrane.

The manuscript largely reports ion permeation experiments with a single cation at a time. To ascertain selectivity, it is necessary to perform mixed ion experiments to assess whether the selectivity observed in single salt experiments also carries over to mixed salt experiments.

In response to the Reviewer's comment, Fig. R4-4 below shows ion transport measurements using equimolar (1 M total) mixtures of different monovalent ion salts for Zr-Ver and Sn-Ver membranes.

For mixtures containing Cs⁺ or K⁺, the interlayer spacing in both membranes collapsed, as expected. Under these conditions, both membranes showed the same selectivity observed with Cs-cation mixtures in the original Fig. 2 (red data points). With hindsight, this was expected. In this case, the one-water-layer-thick channel is expected to display selectivity based only Gibbs hydration energy due to its narrow *d*.

The only mixture that does not contain K⁺ or Cs⁺—and thus does not induce interlayer collapse—is the Na⁺/Li⁺ mixture. In this case, Sn-Ver membranes exhibit a much larger Li⁺ flux than Zr-Ver, reminiscent of the single-salt results, but the Na⁺ flux is about twice that of Li⁺. This represents an intermediate behaviour between the selectivity of Sn-Ver in single-salt experiments, where Li⁺ permeated about six times faster than Na⁺, and that of Zr-Ver membranes, where Na⁺ permeated roughly ten times faster than Li⁺.

We suggest that this latter finding may arise from competition between ions for space within the channels. Na^+ ions are less strongly hydrated than Li^+ , which may lead to a slightly higher affinity to the walls that enhances their concentration in the membrane. Similar competitive effects are commonly reported at mineral–water interfaces, where one ion’s adsorption is altered by the presence of others with different hydration and binding strengths (e.g. S. Yang, et al. *J. Phys. Chem. C* 2019, 124, 1500-1510 and S. Lee, et al. *Geochimica et Cosmo. Acta* 2013, 123, 416-426).

Figure R4-4. Ion flux from mixed salt solutions through Zr-Ver (a) and Sn-Ver (b) membranes, and the (b),(c), Corresponding XRD of membranes in the feed solution.

Was the effect of salt concentration and pH on permeation investigated? In particular, measuring the partitioning of ions, interlayer spacing, and permeability, along with estimating the diffusivity as a function of salt concentration will reveal whether the salt concentration affects the interlayer spacing and ion transport, what kind of adsorption isotherms describe ion uptake into intercalated vermiculite, and whether transport in the dilute limit can explain transport at higher salt concentrations.

Figure R4-5. The effect of salt concentration (a) on the Li ion flux through Zr-Ver membranes. (b), Corresponding XRD of membranes.

Following the Reviewer comment, we investigated the effect of Li concentration on permeation through Zr-Ver membranes. As shown in Fig. R4-5, the Li⁺ flux increased by only by a factor of 2 over the salt concentration range 1mM to 1M, with no detectable change in interlayer spacing.

We also studied the role of pH on ion flux through Zr-Ver. The case of Cs⁺ is particularly relevant because it is the ion with the strongest adhesion to the membrane. Fig. R4-6 shows that the flux is largely insensitive to pH, with a drop in permeability of only a factor of ~2 when the pH changes from 3.5-9.6.

Fig. R4-6. Cs⁺ flux through Zr-Ver as a function of pH. Feed reservoir electrolyte, 1M CsCl.

The hypothesis of entropic effects in the hydration of the intercalating ion on the differences in Li ion transport is very intriguing. However, the reason for the behavior is not pursued apart from correlation to entropy of hydration and mechanical stiffness of the membrane. Does the interlayer spacing change in the presence of Li ions, especially in the case of Sn and Ir intercalation where the membranes are more flexible? Do the observations correlate with the thermodynamic behavior of mixed salt solutions of the intercalating ion and lithium indicating that it may be a bulk effect, or is nanoconfinement essential? What explains the order of magnitude difference in ion partitioning for Sn and Ir intercalated membranes – can it be attributed to interlayer spacing, or are other effects essential? Similarly, does Li partition more favorably than the other monovalent ions in the Sn and Ir intercalated membranes and why (and if not, it must diffuse faster given the data in Fig. 3)? Similarly, temperature dependence to extract the free energy change of partitioning of ions will likely provide additional insights.

The proposed entropic effects explanation was prompted by both our findings and prior literature, as more conventional explanations such as interlayer spacing or bulk thermodynamics did not explain the observed trends. Below we expand on these points.

(1) Interlayer spacing: This was the first possibility we explored. Fig. 3b in the original manuscript shows that all four membranes immersed in Li⁺ electrolyte exhibit the same interlayer spacing ($\approx 14.8 \text{ \AA}$), ruling out channel height differences as the source of the selectivity.

(2) Partitioning vs. diffusion: To unpick this unexpected finding, we measured Li⁺ concentration in the membranes after permeation measurements using mass spectrometry (Supplementary Fig. 11). The data showed that Li⁺ concentration in Ir-Ver and Sn-Ver is an order of magnitude higher than in Zr-Ver and La-Ver (Supplementary Fig. 11). This revealed that partitioning, rather than

diffusion, was the dominant factor. Beyond this, and following a comment from Reviewer #1, we have now directly probed the diffusion in the membranes using electrochemical impedance spectroscopy. This revealed that the diffusion constants of Li changes by only about ~20% for the different membranes (Fig. R1-1, page 1 in this file and Supplementary Fig. 12 in revised manuscript). This confirms that partitioning, rather than diffusion, explain the differences in permeability.

Figure R4-7. temperature-dependent measurements of ion partitioning in the Zr-Ver membrane

Following the Reviewer comment, we now include new temperature-dependent measurements of Li^+ ion partitioning. As shown in Figure R4-7, the temperature dependence is very weak.

(3) Thermodynamics of bulk solutions: Following the Reviewer comment, we also considered thermodynamic behaviour of bulk solutions. From Debye-Hickel theory, the presence of highly charged intercalated ions, such as Ir^{4+} and Zr^{4+} , should result in reduced ion activity coefficients for the permeating ions. However, since both Ir^{4+} and Zr^{4+} have the same charge, this cannot explain our observations.

(4) Proposed explanation: Having ruled out channel height and ion activity as explanations, we turned to the biological literature, where channel elasticity is known to influence ion selectivity by altering the Gibbs energy of ions inside elastic versus rigid channels (Laio & Torre, Biophys. J. 1999). Similar coupling between mechanics and ion exchange has also been observed in clays (Whittaker et al., PNAS 2019). Motivated by this, we measured the Young's modulus of our membranes and found two distinct groups: stiff (Zr-Ver, La-Ver) and soft (Ir-Ver, Sn-Ver), with moduli differing by an order of magnitude. The stiff membranes showed classical selectivity ($\text{Cs}^+ > \text{K}^+ > \text{Na}^+ \gtrsim \text{Li}^+$), while the soft ones showed the reverse ($\text{Li}^+ > \text{Na}^+ \approx \text{Cs}^+ \approx \text{K}^+$). Because the only difference between the two groups is the intercalated ion, we attributed the changes in stiffness and selectivity to these ions. Marcus' seminal work explains this link: ions with low hydration entropy (e.g. Zr^{4+}) lead to more ordered and thus stiffer water, while those with higher entropy (e.g. Sn^{4+}) lead to more disordered water. Such effects, demonstrated in bulk, should be amplified under confinement, where water is more structured and resistant to deformation (as shown in AFM and surface force studies, Refs. 47, 49). This amplification should make confined water especially sensitive to the influence of intercalated ions, providing a natural explanation for the stiffness–selectivity correlation we observe.

Response to comments from Reviewer #1

The authors have satisfactorily addressed the issues raised during the initial review. This work is now suitable for publication.

We are grateful to the Reviewer for this recommendation.

Response to comments from Reviewer #2

The authors have addressed my comments properly, and I would like to recommend the publication of this manuscript in Nature Communications.

We thank the Reviewer for this assessment of our work.

Response to comments from Reviewer #3

The authors have addressed most of my concerns. I have two minor comments that I would like the authors to further clarify before this work is published:

We thank the Reviewer for this assessment of our revised manuscript. We have done our best to clarify the points below.

1. Monovalent ion permeation with varying channel widths

Referring to my previous Comment 3, I remain surprised by the relatively unaffected permeation of monovalent ions with different channel widths (Fig. 2e). I understand that ion permeation is governed by the thermodynamic partitioning step, which depends on ion dehydration and ion–pore interactions. A smaller pore is typically expected to impose a higher dehydration penalty. At the same time, ions with fewer surrounding water molecules should interact more strongly with the pore walls, partially compensating for this penalty. The authors suggest that the observed unaffected monovalent permeation arises from a balance: the additional dehydration penalty is approximately offset by stronger ion–pore interactions, resulting in a similar overall free energy change for both channel sizes. My question is: is this balance merely coincidental in this case, or does it represent a more universal principle?

Our understanding is that this balance is coincidental. We observe it only for monovalent ions, which are relatively weakly hydrated. For multivalent ions, which are more strongly hydrated, we observe much sharper rejection as function of hydration energy in narrow channels (Fig. 2, red dots). Thus, since this balance is not expected in general, we regard it as coincidental—although perhaps not entirely unexpected with hindsight, given these minerals' well-documented affinity for monovalent ions.

In search for a more universal principle behind this observation, perhaps it may be helpful to note that a key principle of statistical mechanics is that distinct microscopic configurations of a system can yield the same ΔG . Since ΔG determines ion concentration and permeability in these membranes, different configurations of the system can yield the same permeability.

2. Relation between elasticity and selectivity

Referring to my previous Comment 9, I appreciate the improved discussion of the mechanism, but I still find the link between membrane elasticity and ion selectivity insufficiently explicit. The current revision convincingly shows that pore elasticity influences ion selectivity, and that incorporating different non-exchangeable ions alters the elasticity of the membrane. However, it remains unclear, at least to me, and potentially to readers, why Li^+ permeates most slowly through membranes with higher Young's modulus (Zr-Ver and La-Ver) and most rapidly through those with lower Young's modulus (Sn-Ver and Ir-Ver). A clearer mechanistic explanation of this trend would strengthen the interpretation.

Our mass spectrometry data show that Li^+ permeates more slowly in membranes with higher stiffness because its equilibrium concentration inside the membrane is lower. This difference reflects the Gibbs energy of Li^+ in the membrane, which includes an elastic contribution. In the simplest approximation, this energy is given by $H_e(\bar{x}) = (1/2) k(\bar{x}) [d(\bar{x}) - d_o(\bar{x})]^2$, where $k(\bar{x})$ is the stiffness as a function of position in the channel, \bar{x} , and $d_o(\bar{x})$ is the equilibrium channel width. Li^+ has the largest hydration shell amongst the monovalent ions. As a result, its associated term $d(\bar{x}) - d_o(\bar{x})$ in $H_e(\bar{x})$ should display larger differences than other ions and should thus be more strongly affected by a change in k . Following the Reviewer comment, we have expanded on this point in the revised manuscript, in the first and second paragraphs in page 7.

Response to comments from Reviewer #4

The authors have addressed all the reviewer comments satisfactorily and the reviewer recommends publication without further revision.

We thank the Reviewer for their support of our work.